# Towards Memorization Estimation: Fast, Formal and Free

**Deepak Ravikumar** [1]  **Efstathia Soufleri** [1]  **Abolfazl Hashemi** [1]  **Kaushik Roy** [1]

## Abstract

Deep learning has become the de facto approach in nearly all learning tasks. It has been observed that deep models tend to memorize and sometimes overfit data, which can lead to compromises in performance, privacy, and other critical metrics. In this paper, we explore the theoretical foundations that connect memorization to sample loss, focusing on learning dynamics to understand what and how deep models memorize. To this end, we introduce a novel proxy for memorization: Cumulative Sample Loss (CSL). CSL represents the accumulated loss of a sample throughout the training process. CSL exhibits remarkable similarity to stability-based memorization, as evidenced by considerably high cosine similarity scores. We delve into the theory behind these results, demonstrating that low CSL leads to nontrivial bounds on the extent of stability-based memorization and learning time. The proposed proxy, CSL, is four orders of magnitude less computationally expensive than the stability-based method and can be obtained with zero additional overhead during training. We demonstrate the practical utility of the proposed proxy in identifying mislabeled samples and detecting duplicates where our metric achieves state-of-the-art performance.

## 1. Introduction

Deep learning has become the de facto standard for almost all machine learning tasks from image (Ho et al., 2020) and text generation (Radford et al., 2019) to classification (Krizhevsky et al., 2009; Soufleri et al., 2024a) and reinforcement learning (Shakya et al., 2023). While they have been extremely successful, they tend to memorize and over-

[1]Department of Electrical and Computer Engineering, Purdue University, West Lafayette, U.S.A. Correspondence to: Deepak Ravikumar <dravikum@purdue.edu>.

*Proceedings of the 42nd International Conference on Machine Learning*, Vancouver, Canada. PMLR 267, 2025. Copyright 2025 by the author(s).

[2]Link to the implementation: https://github.com/DeepakTatachar/CSL-Mem

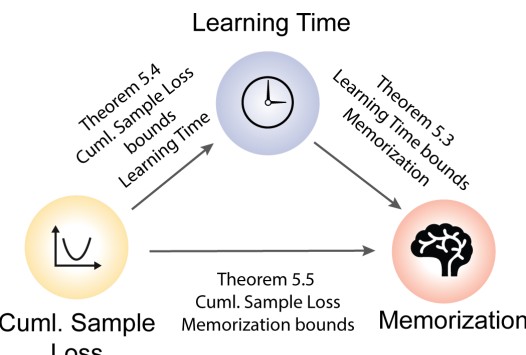

*Figure 1.* Visualizing our contributions linking cumulative sample loss to memorization and learning time.

fit to the training data. While some memorization is indeed needed to obtain generalization (Feldman, 2020), these deep models can also memorize totally random images (Zhang et al., 2017). Thus to understand memorization, researchers have put in significant effort (Zhang et al., 2017; Arpit et al., 2017; Carlini et al., 2019a; Feldman & Vondrak, 2019; Feldman & Zhang, 2020; Feldman, 2020). Such focus is crucial due to the broad implications of memorization for multiple connected areas, including generalization (Zhang et al., 2021; Brown et al., 2021), noisy learning (Liu et al., 2020), identifying mislabeled examples (Maini et al., 2022), recognizing rare and challenging instances (Carlini et al., 2019a), ensuring privacy (Feldman, 2020), and addressing risks from membership inference attacks (Shokri et al., 2017; Carlini et al., 2022; Ravikumar et al., 2024b).

Many approaches to study memorization have been proposed (Carlini et al., 2019a; Jiang et al., 2021; Feldman, 2020). Notably, the stability-based measure proposed by Feldman (2020) measures the change in expected output probability when the sample under investigation is removed from the training dataset. This measure offers a robust theoretical framework for understanding memorization, which was subsequently validated empirically for deep neural networks (Feldman & Zhang, 2020). However, this approach is impractical for most applications due to its high computational cost. Recent literature has introduced other proxies for memorization, such as learning time (Jiang et al., 2021), adversarial distance (Del Grosso et al., 2022), model confidence (Carlini et al., 2019b), and input loss curvature (Garg et al., 2024; Ravikumar et al., 2024a). While these proxies

have been successful in understanding the memorization behavior of neural networks, most fail to capture certain properties of memorization such as bi-modality (Lukasik et al., 2023). Thus, establishing a strong theoretical foundation of memorization and its proxies is of critical importance.

While prior work (Ravikumar et al., 2024a) has investigated the properties of the loss function, such as input curvature post-training and its connection to memorization, we establish a theoretical framework that explains how learning dynamics drive the similarity between loss, memorization and learning time. We propose a new proxy for memorization: Cumulative Sample Loss (CSL) to capture information from training dynamics. CSL represents the loss of a sample accumulated over the entire training process. The proposed CSL proxy is *4 orders of magnitude* less computationally expensive than stability-based (Feldman & Zhang, 2020) memorization and $\approx$ *14$\times$* less expensive than input loss curvature (Garg et al., 2024). It is important to note that the *14$\times$* estimate is conservative. This is because CSL can be obtained for free during training, making the computational benefits even greater than these numbers suggest.

We validate our theory with experiments and show that the proposed proxy has a very high cosine similarity with the memorization score from (Feldman & Zhang, 2020). Further, we show that CSL can be used to identify duplicates and mislabeled examples; notably, the adoption of our proposed proxy leads to achieving state-of-the-art performance in these applications. In summary, our contributions are:

- We present a new theoretical framework that links learning dynamics (e.g. sample learning time) and memorization, to cumulative sample loss as visualized in Figure 1.

- We propose a new memorization proxy: Cumulative Sample Loss (CSL), which demonstrates very high similarity to stability-based memorization methods and is significantly more computationally efficient, offering a reduction in computational cost by several orders of magnitude.

- We validate our theory through experiments on deep vision models, demonstrating the efficacy of CSL as a strong memorization proxy.

- We showcase the practical applications of our proxy in identifying mislabeled examples and duplicates in datasets, achieving state-of-the-art performance in these tasks.

## 2. Notation

We denote distributions using bold capital letters $\mathbf{V}$, random variables sampled from them as italic small letters $v$

for scalars, $\vec{v}$ for vectors, and capital letters $V$ for matrices. All constants are represented using Greek symbols (with two exceptions $L$ and $T_{\max}$). For simplicity and compactness, we ignore the notation when vectors are in the subscript, for example $\nabla_w = \nabla_{\vec{w}}$. Consider a learning problem, where the task is learning the mapping $f : \vec{x} \mapsto y$ where $\vec{x} \sim \mathbf{X} \in \mathbb{R}^n$ and $y \sim \mathbf{Y} \mid \mathbf{X} \in \mathbb{R}$. A dataset $S = (\vec{z}_1, \vec{z}_2, \ldots, \vec{z}_m) \sim \mathbf{Z}^m$ consists of $m$ samples, where each sample $\vec{z}_i = (\vec{x}_i, y_i) \sim \mathbf{Z}$. We also use a leave one out set which the dataset $S$ with the $i^{th}$ sample removed denoted by $S^{\backslash i} = (\vec{z}_1, \ldots, \vec{z}_{i-1}, \vec{z}_{i+1}, \ldots, \vec{z}_m)$. We use $g_S^p \sim \mathbf{G}_S$ to denote the function learnt by the neural network by the application of a possibly random training algorithm $\mathcal{A}$, on the dataset $S$ where $p \sim \mathbf{P}$ denotes the randomness of the algorithm. Let the row vector $\vec{w}_t = [\vec{w}_t^{(1)}, \vec{w}_t^{(2)}, \cdots, \vec{w}_t^{(q)}] \sim \mathbf{W}$ denote the weights of a $q$-layered network at iteration $t$, where $\vec{w}_t^{(k)} = \begin{bmatrix} w_{t,1,1}^{(k)} & w_{t,1,2}^{(k)} & \cdots & w_{t,d_k,d_{k-1}}^{(k)} \end{bmatrix}$ is a flattened row vector representing the weights of the $k^{th}$ layer at iteration $t$ with input dimension $d_{k-1}$ and output dimension $d_k$. Now, consider a single data sample $\vec{x}_i = \begin{bmatrix} x_{i1} & x_{i2} & \cdots & x_{in} \end{bmatrix}^\top$ represented as a column vector. Then, we represent a mini-batch with $b$ examples as a matrix $X_b = \begin{bmatrix} \vec{x}_1 & \vec{x}_2 & \cdots & \vec{x}_b \end{bmatrix}$, whose corresponding labels are denoted as $Y_b = [y_1 \cdots y_b]$. Thus a mini-batch is denoted as $Z_b = (X_b, Y_b)$. A cost function $c : \text{Range}(\mathbf{Y}) \times \text{Range}(\mathbf{Y}) \to \mathbb{R}^+$, is used to evaluate the performance of the model. The cost at a sample $\vec{z}_i$ is referred to as the loss $\ell$ evaluated at $\vec{z}_i$, defined as $\ell(g, \vec{z}_i) = c(g(\vec{x}_i), y_i)$. In some cases, the loss function may involve weights or parameters $\vec{w}_t$, such as when the model is parameterized by weights. In such cases, we may write the loss function as $\ell(\vec{w}_t, \vec{z}_i)$. When working with a batch of data, the loss function can be extended to a batch of samples $Z_b = (X_b, Y_b)$. In this case, the loss function may be written as $\ell(\vec{w}_t, Z_b)$ where the loss for the batch is typically the average or sum of the losses for the individual samples within the batch. Typically, we are interested in the loss of $g$ over the entire data distribution, called the population risk, which is defined as $R(g) = \mathbb{E}_z[\ell(g, \vec{z})]$. Since the data distribution $\mathbf{Z}$ is generally unknown, we instead evaluate the empirical risk as follows $R_{\text{emp}}(g, S) = \frac{1}{m} \sum_{i=1}^m \ell(g, \vec{z}_i), \vec{z}_i \in S$.

## 3. Background

**Memorization** of the $i^{th}$ element $\vec{z}_i = (\vec{x}_i, y_i)$ in the dataset $S$ by an algorithm $\mathcal{A}$ is as:

$$\text{mem}(S, \vec{z}_i) = \Pr_p[g_S^p(\vec{x}_i) = y_i] - \Pr_p[g_{S^{\backslash i}}^p(\vec{x}_i) = y_i] \quad (1)$$

where the probability is taken over the randomness of the algorithm $\mathcal{A}$. This definition of memorization is the from Feldman (2020).

**$\rho$-Lipschitz Gradient.** The gradient of the loss function $\ell$

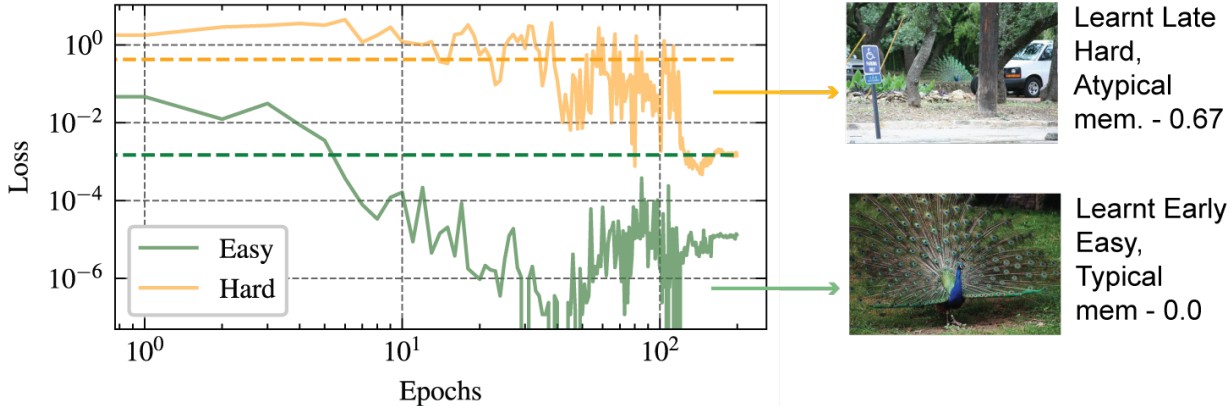

*Figure 2.* Visualizing peacock class learning in ImageNet. Average loss is shown for easy and hard-to-learn peacocks. The dashed line represents average loss, while solid lines show actual loss. Easy images are less memorized, while hard images are memorized more.

is said to be $\rho$-Lipschitz on Range($\mathbf{W}$) if, for all $\vec{w}_1, \vec{w}_2 \in$ Range($\mathbf{W}$), there exists a constant $\rho > 0$ such that:

$$\|\nabla_{w_1}\ell(\vec{w}_1) - \nabla_{w_2}\ell(\vec{w}_2)\| \leq \rho\|\vec{w}_1 - \vec{w}_2\| \qquad (2)$$

**Bounded Gradient.** Suppose that for each iteration $t$, the expected euclidean norm of the gradient with respect to the parameters $\vec{w}_t^k$ is bounded by a constant $\Gamma^2$, i.e.,

$$\mathbb{E}_t\left[\|\widetilde{\nabla}_{w_t^k}\ell(\vec{w}_t^k)\|_2^2\right] \leq \Gamma^2, \qquad (3)$$

**Uniform Stability** of a randomized algorithm $\mathcal{A}$ for some $\beta > 0$ is defined as in Kearns & Ron (1997):

$$\left|\ell(g_S^p, \vec{z}) - \ell(g_{S\setminus i}^p, \vec{z})\right| \leq \beta, \quad \forall i \in \{1, \cdots, m\} \qquad (4)$$

**L-Bounded Loss.** We say that loss a loss function is L-bounded if it satisfies $0 \leq \ell \leq L$.

## 4. Related Work

**Memorization** in deep neural networks has gained attention, with recent works improving our understanding of its mechanisms and implications (Zhang et al., 2017; Arpit et al., 2017; Carlini et al., 2019a; Feldman & Vondrak, 2019; Feldman, 2020; Feldman & Zhang, 2020; Lukasik et al., 2023; Garg et al., 2024; Ravikumar et al., 2024a). This research is driven by the need to understand generalization (Zhang et al., 2017; Brown et al., 2021; Zhang et al., 2021; Kaplun et al., 2022; Bayat et al., 2025), identify mislabeled examples (Pleiss et al., 2020; Maini et al., 2022), and detect out-of-distribution or rare sub-populations (Carlini et al., 2019a; Ravikumar et al., 2023; Pezeshki et al., 2023). Additionally, memorization impacts robustness (Shokri et al., 2017; Carlini et al., 2022), unlearning (Kurmanji et al., 2023; Kodge et al., 2024) and privacy (Feldman, 2020; Soufleri et al., 2024b).

**Privacy** is often tested using membership inference attacks (MIA), which tests whether a particular sample was used in training (Shokri et al., 2017; Carlini et al., 2022; Ravikumar et al., 2024b). Learning dynamics has been used in this context to build stronger privacy tests. Liu et al. (2022) leverage loss trajectories and distilled models to improve MIA, while Li et al. (2024) propose SeqMIA, an attention-based recurrent architecture that exploits intermediate model states. Both approaches demonstrate how learning dynamics can reveal training-set membership but at the cost of increased computational overhead. Additionally, Nasr et al. (2021) highlights how mean loss trajectories can reveal privacy leakage under differentially private training establishing a lower bound on leakage identification.

**Learning Dynamics**. Beyond privacy, learning dynamics have been studied as proxies for memorization. Mangalam & Prabhu (2019) showed simpler examples are learned first while mislabeled or difficult samples may be repeatedly forgotten or relearned (Toneva et al., 2019; Pleiss et al., 2020; Maini et al., 2022). Jagielski et al. (2023) explored how memorization fades when examples are removed. Carlini et al. (2019a) combine multiple metrics to understand data retention, and Jiang et al. (2021) introduce the C-score to capture hard examples. Other works have proposed loss-sensitivity (Arpit et al., 2017), and sharpness of the loss function (Krueger et al., 2017) as memorization signals. More recently, Garg et al. (2024) used input loss curvature as a proxy for stability-based memorization (Feldman, 2020), supported by theoretical analysis from Ravikumar et al. (2024a). In contrast, this paper investigates sample loss over training and its formal link to learning time and memorization.

## 5. Learning Dynamics and Memorization

### 5.1. Proposed Memorization Proxy

To build intuition, let us explore the loss progression of two samples, namely, an "easy" and a "hard" example, both from the same class (peacock) in the ImageNet dataset (Rus-

sakovsky et al., 2015), as illustrated in Figure 2. In this context, loss refers to the per-sample cross-entropy loss, which tracks how well the model predicts a specific example at each epoch of training.

In Figure 2, let's observe the behavior of an easy sample, represented by the loss visualized in green. This sample is learned early in the process. The loss follows a simple pattern: it starts high, quickly drops, and stays low for the rest of the training. The hard sample, on the other hand, behaves differently. Its loss remains high for a much longer period before eventually dropping, indicating that it is learned much later. This contrast in loss dynamics' patterns evidently shows how the cumulative per-sample loss throughout training can distinguish between easy and hard examples with precision. Figure 2 also illustrates that using thresholds (i.e. loss threshold) to define learning time and forgetting time, is less effective. This is because learning is a noisy process, i.e. a sample might be learnt in one epoch, unlearnt in the next. Thus traditional proxies which rely on thresholds to determine when a sample is learnt, may fall short of distinguishing between easy and hard examples. To overcome this noise, we propose using cumulative sample loss, CSL as more reliable proxy. The proposed proxy CSL of a sample $\vec{z}_i$ can be formally defined as:

$$\text{CSL}(\vec{z}_i) = \sum_{t=0}^{T_{max}-1} \ell(\vec{w}_t, \vec{z}_i) \quad (5)$$

where $T_{max}$ is the total number of iterations of SGD. This smooths out the noise from fluctuations during learning. As we will demonstrate, hard examples tend to be memorized by the model, while easy examples are likely generalized; the proposed cumulative proxy, tracked throughout the training process, is key to capturing this correlation.

### 5.2. Learning Condition and Learning Time

To formalize this intuition, we introduce the concept of the *sample learning condition*. In optimization theory, the necessary condition for optimality for an unconstrained problem is typically expressed as $\nabla_{w_t} \ell(\vec{w}_t) = 0$. In the case of optimizers like gradient descent or its extensions, convergence is typically characterized in terms of gradient norm given as $\|\nabla_{w_t} \ell(\vec{w}_t)\|_2 \leq \tau$, where $\ell$ is the function to minimize and $\tau$ denotes an arbitrarily small threshold. Thus, as a extension of this perspective, we define the *sample learning condition* as:

$$\mathbb{E}_R \left[ \|\nabla_{x_i} \ell(\vec{w}_R)\|_2^2 \right] \approx \frac{1}{T} \sum_{t=0}^{T-1} \|\nabla_{x_i} \ell(\vec{w}_t)\|_2^2 \leq \tau, \quad (6)$$

where *the gradient is with respect to the input $\vec{x}_i$*. We interpret Equation 6 as follows: A sample is considered learnt if

the *average per iteration sample gradient norm* falls below a threshold $\tau$. Note that the approximation is justified by the law of large numbers. Using this idea we formally define the learning time $T_{z_i}$ of a sample $\vec{z}_i = (\vec{x}_i, y_i)$ as the smallest time $T$ at which the sample learning condition is satisfied. Formally:

$$T_{z_i} = \min_{T} \left\{ T : \frac{1}{T} \sum_{t=0}^{T-1} \|\nabla_{x_i} \ell(\vec{w}_t)\|_2^2 \leq \tau \right\} \quad (7)$$

where $\tau > 0$ is a predefined threshold that determines the acceptable magnitude of the average gradient norm. While the definitions are non-intuitive, as we will demonstrate, this formulation of sample learning condition and learning time simplifies the mathematical expressions and analyses.

### 5.3. Input Gradient Convergence

In this section, we analyze the evolution of input-space gradients during training. Specifically, we demonstrate that the norm of the gradient with respect to the weights bounds the norm of the gradient with respect to the input, ensuring convergence of the input gradient. Consequently, the sample learning condition defined in Eq. 6 also converges. To establish this, we examine the gradient of the loss function with respect to a sample, and relate it to the gradient with respect to the network weights.

**Lemma 5.1** (Input gradient norm is bound by weight gradient norm). *For any neural network, given a mini-batch of inputs $Z_b = (X_b, Y_b)$, the Frobenius norm of the gradient of the loss $\ell$ with respect to the input is bounded by the norm of the gradient with respect to the network's weights $\vec{w}_t$. Specifically:*

$$\|\nabla_{X_b} \ell(\vec{w}_t, Z_b)\|_F \leq k_g \|\nabla_{w_t} \ell(\vec{w}_t, Z_b)\|_F \quad (8)$$

*where $k_g = \frac{\|W_t^{(1)}\|_F \|(X_b^\top)^+\|_F}{s_P}$ and $s_P$ denotes the smallest singular value of $P = X_b^\top (X_b^\top)^+$, where $^+$ denotes pseudo-inverse.*

**Sketch of Proof.** The proof utilizes the chain rule to compute the gradients of the loss with respect to the inputs and the with respect to first-layer weights, establishing the relation between the two. By leveraging the Frobenius norm, the proof establishes an upper bound on the gradient with respect to the inputs in terms of the gradient with respect to the weights. The proof is available in Appendix C.1.

For convenience we group a set of assumptions as *SGD Convergence Assumptions*. The convergence of SGD in gradient norm holds under the $\rho$-Lipschitz continuity of the loss function (Eq. 2), with a bounded variance norm (Eq. 16) and an unbiased gradient estimator.

**Theorem 5.2** (Convergence in input gradient norm). *Under the SGD convergence assumptions, after $T$ iterations of*

*SGD with a learning rate $\eta = 1/\sqrt{T}$, the sample learning condition converges as follows:*

$$\frac{1}{T}\sum_{t=0}^{T-1}\|\nabla_{x_i}\ell(\vec{w}_t)\|_2^2 = \mathcal{O}\left(1/\sqrt{T}\right) \qquad (9)$$

**Sketch of Proof.** The proof starts with SGD convergence analysis. Subsequently, using Lemma 5.1, the proof relates the weight gradient to the input gradient through a scaling factor that depends on $k_g$. By de-telescoping the result over iterations, it follows that the cumulative input gradient is bounded by the cumulative loss decrease and a term depending on the learning rate, Lipschitz constant and iterations $T$. Using $\eta = 1/\sqrt{T}$ we see that the cumulative input gradient. converges in $\mathcal{O}(1/\sqrt{T})$. Full proof is in Appendix D.1.

**Discussion.** The theorem establishes the convergence of the sample gradient norm when using SGD. It shows that the input gradient converges in a manner similar to the weight gradient. The difference being the upper bound has a dependence on the scaling factor $\kappa_g$ from Lemma 5.1.

**Takeaways.** *(1) The big takeaway is that when we train a model, even though we don't optimize in the input space, we do indirectly optimize it. (2) Sample learning condition (Eq. 6) is guaranteed to converge.*

### 5.4. Learning Time, CSL and Memorization

**Theorem 5.3** (Learning Time bounds Memorization)**.** *Under the assumptions of SGD convergence, $\beta$-stability and $L$-bounded loss, there exists a $\kappa_T > 0$ such that the expected learning time for a sample $\vec{z}_i$ bounds expected memorization score, i.e.*

$$\mathbb{E}_{z_i}\left[\operatorname{mem}(\vec{z}_i)\right] \leq \kappa_T \,\mathbb{E}_{z_i,p}\left[T_{z_i}\right] + \beta/L, \qquad (10)$$

*where $p$ denotes the randomness of the algorithm.*

**Sketch of Proof.** The key idea is to leverage the Theorem 5.2 to derive a closed-form expression for learning time. The proof begins by relating $T_{z_i}$ to the cumulative reduction in loss over training steps. The loss dynamics are decomposed into two terms: the memorization score, and a residual term capturing the generalization gap. By analyzing these components and taking expectations over the data and randomness of the training process, it is shown that the expected learning time scales with the memorization score. The full proof is available in Appendix D.2.

**Interpreting Theory.** Theorem 5.3 intuitively states that the expected memorization of a group of samples is proportional to their expected learning times. To make this clearer, consider a subset $U(T) \subseteq S$, where $U(T) = \{\vec{z}_i : T_{z_i} \leq T\}$, meaning $U(T)$ contains all samples that are learned within a time threshold $T$. From the relationship between memorization and learning time established in Equation 10, it

follows that the expected memorization of samples in this subset is bounded by their learning times. Given that

$$\kappa_T = \frac{\tau\eta}{\kappa_m L} - \frac{\eta^2 \rho \Gamma^2}{2L} \geq 0$$

where $\kappa_m = \max_{t\in 1,\cdots,T}(\kappa_g^t)^2$, it can be assumed that $\kappa_T$ is constant across different subsets $U(T)$. Thus, we can state that for any sample $z_i \in U(T)$, the expected memorization $\mathbb{E}_{z_i}[\operatorname{mem}(z_i)]$ follows

$$\mathbb{E}_{z_i\in U(T)}[\operatorname{mem}(z_i)] \leq \kappa_T T + \beta/L$$

*In simpler terms, this means that if a group of samples is learned early in the training process (i.e., they have smaller learning times), their expected memorization will also be lower compared to samples that are learned much later during training.* This provides an intuitive link between learning time and the extent of memorization, suggesting that samples requiring more training time are more likely to be memorized. Additionally, it can be demonstrated that Theorem 5.3 holds true for cross-entropy loss, by setting $L = 1$ (see proof in Appendix D.2.1). Further, under reasonable generalization assumption $\mathbb{E}_{t,z_i,p}\left[\ell(\vec{w}_0, \vec{z}_i) - \ell^{\backslash z_i}(\vec{w}^*, \vec{z}_i)\right] \geq 0$, the requirement for stability can be dropped (see discussion in Appendix D.2.2), leading to:

$$\mathbb{E}_{z_i}\left[\operatorname{mem}(\vec{z}_i)\right] \leq \kappa_T \,\mathbb{E}_{z_i,p}\left[T_{z_i}\right] \qquad (11)$$

**Theorem 5.4** (Cumulative loss bounds learning time)**.** *Let the assumptions for SGD convergence hold. Then, for any $L$-bounded loss, the learning time $T_{z_i}$ for any sample $\vec{z}_i \in S$ there exist a $\kappa_T > 0$ such that:*

$$\kappa_T \,\mathbb{E}_{z_i}\left[T_{z_i}\right] \leq \frac{\mathbb{E}_{z_i}\left[\operatorname{CSL}(\vec{z}_i)\right] - \xi}{L} \qquad (12)$$

*where $\xi$ is an offset to scale CSL into the correct range.*

**Sketch of Proof.** The proof begins by leveraging the input gradient convergence result (Theorem 5.2), which allows us to derive an expression for the learning threshold $\tau$. By rearranging this result, we establish a direct connection between $T_{z_i}$ and CSL, demonstrating that the learning time scales with the cumulative sample loss. To refine this relationship, we define $\xi$ as an offset representing the aggregate lower bound of the cumulative loss. Finally, by combining these results, we show that the expected learning time is proportional to the expected cumulative sample loss. The proof is detailed in Appendix D.3. Additionally, this theorem holds for any unbounded loss with by setting $L = 1$ (see proof in Appendix D.3.1).

**Interpreting Theory.** The theorem highlights that the learning time for a sample is directly tied to its Cumulative Sample Loss (CSL), which quantifies the total loss over time. This makes it possible to compare different subsets

of samples based on their CSL and understand how it affects their learning time. $\xi$ is an offset such that CSL is in the correct range, this is more clear when considering the result for memorization (see Theorem 5.5), we observe that $\xi$ scales CSL so that it lies in the range $\approx (0, 1)$ when $\beta \approx 0$ (see more detailed discussion in Appendix D.3.2). To build intuition, consider a subset $U(C) \subseteq S$, defined as $U(C) = \{\vec{z}_i : \text{CSL}(\vec{z}_i) \leq C\}$. According to Theorem 5.4, which applies to any subset, the following holds:

$$\mathbb{E}_{z_i \in U(C)}[T_{z_i}] \leq \frac{C - \xi'}{L}$$

Here $\xi'$ is the minimum $\xi$ across various subsets, thus $\xi'$ is the same across all subsets. Hence, we can compare subsets by their CSL: *subsets with lower CSL will have shorter learning times. In simpler terms, if one group of samples has a lower CSL than another, they are expected to be learnt earlier during training.* If the bounds are accurate, this relationship is close to linear, meaning lower CSL consistently leads to shorter learning times.

**Theorem 5.5** (Cumulative Sample Loss bounds Memorization)**.** *Assume the loss function is $L$-bounded, and the assumptions for the convergence of SGD and $\beta$-stability hold. Then, the expected memorization score of any sample $\vec{z}_i \in S$ satisfies the following inequality:*

$$\mathbb{E}_{z_i}[\text{mem}(\vec{z}_i)] \leq \frac{\mathbb{E}_{z_i,p}[\text{CSL}(\vec{z}_i)] + \beta - \xi}{L} \leq 1 + \frac{\beta}{L} \tag{13}$$

*where $p$ denotes the randomness of the algorithm.*

**Sketch of proof.** The proof connects memorization to CSL using Theorems 5.3 and 5.4. By taking the expectation over the algorithm's randomness in Theorem 5.4 and using it to bound Theorem 5.3, the result follows. The proof is provided in Appendix D.4.

**Interpreting Theory.** Consider a subset $U(C) \subseteq S$ where $U(C) = \{\vec{z}_i : \text{CSL}(\vec{z}_i) \leq C\}$. For any two such sets, the set with a higher CSL is likely to contain more memorized examples. *In simpler terms, if one group of samples has a lower CSL than another, it is expected to be less memorized and learned earlier during training.* If the bounds from this theorem are tight, we can expect a linear relationship between learning time and memorization. Under a reasonable generalization assumption, similar to Eq. 11, memorization is shown to be bounded by normalized CSL, which is upper-bounded by 1. This demonstrates that the CSL bound is non-trivial. Additionally, for cross-entropy loss the following holds true (details in Appendix D.4.1):

$$\mathbb{E}_{z_i}[\text{mem}(\vec{z}_i)] \leq \mathbb{E}_{z_i,p}[\text{CSL}(\vec{z}_i)] + \beta - \xi \tag{14}$$

**Remark on Assumptions.** We briefly and qualitatively evaluate the practicality of our assumptions. It has been established that SGD methods are uniformly stable (Hardt et al., 2016), supporting the plausibility of our assumptions on stability (Eq. 4). Virmaux & Scaman (2018) provides a general upper bound for the Lipschitz constant of any differentiable deep learning model, validating the Lipschitz continuity assumption in the context of deep models. The assumptions of an unbiased gradient estimator, along with bounded gradient norm, are widely used in the optimization literature (Ghadimi & Lan, 2013; Aketi et al., 2024), making these assumptions reasonable. In practice, loss functions are often upper-bounded. Additionally, our results apply to cross-entropy with minor changes as specified.

**Key Theory Takeaways.** If the theory bounds are tight, we expect the following: (1) learning time to exhibit a linear relationship with CSL, (2) stability-based memorization to also follow a linear relationship with CSL, and (3) a linear relationship between learning time and memorization.

# 6. Experiments

## 6.1. Validating Theory

In this section, we conduct experiments to empirically validate the theoretical relationships established in the paper. Specifically, we investigate the following connections: (1) the relationship between learning time and our proxy CSL (Cumulative Sample Loss), (2) the relationship between memorization and learning time (3) the relation between CSL and memorization.

**Experiment.** We trained a ResNet18 model on the CIFAR-100 and ImageNet datasets, measuring learning time and CSL for each training sample. For memorization scores, we used precomputed stability-based scores from Feldman & Zhang (2020). To analyze the relationship between the CSL, learning time and memorization, we created a binned scatter plot. Since the theorems apply to groups of samples (i.e. under expectation), we grouped data by the x-axis metric and calculated average scores for each bin. For instance, in Figure 3, samples are grouped by memorization scores on the x-axis, and the average learning time for each bin is plotted to generate the scatter plot (additional details are provided in Appendix B.4).

**Results.** The results are visualized in Figures 3, 4, 5, 6, 7, and 8. Figures 3 and 6 illustrate the relationship between learning time and memorization for CIFAR-100 and ImageNet respectively, validating Theorem 5.3. Figures 4 and 7 plot learning time versus CSL for CIFAR-100 and ImageNet, respectively, supporting Theorem 5.4. Figures 5 and 8 compare the memorization score from Feldman & Zhang (2020) with CSL for CIFAR-100 and ImageNet, respectively, validating Theorem 5.5.

**Takeaways.** Theorems 5.3–5.5 establish bounds under expectation over groups of samples, describing the relation-

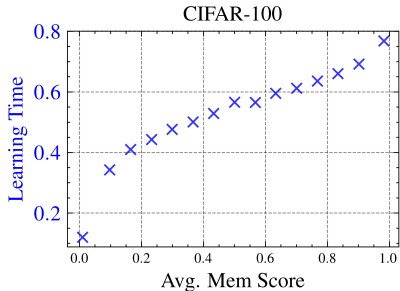

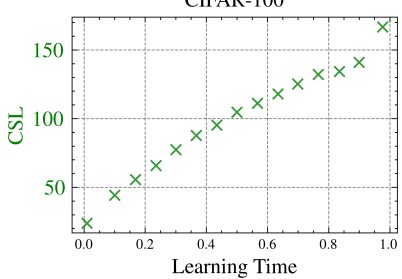

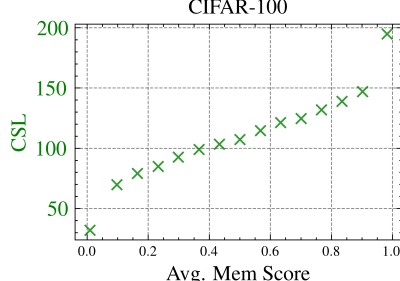

*Figure 3.* Learning Time vs Memorization score on CIFAR-100 dataset.

*Figure 4.* Learning time vs. CSL on the CIFAR-100 dataset.

*Figure 5.* Memorization score vs CSL on CIFAR-100 dataset.

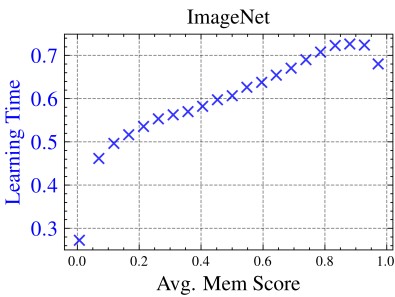

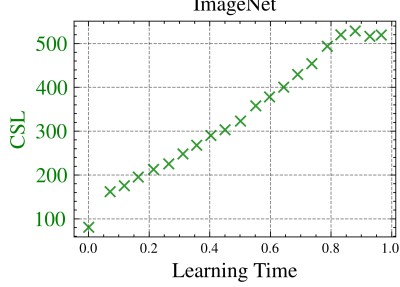

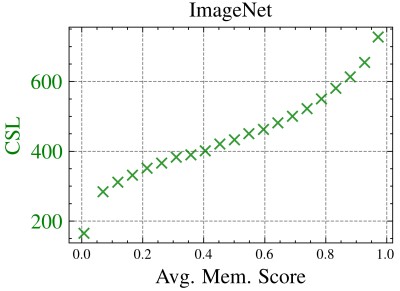

*Figure 6.* Learning Time vs Memorization score on ImageNet dataset.

*Figure 7.* Learning time vs. CSL on the ImageNet dataset.

*Figure 8.* Memorization score vs CSL on ImageNet dataset.

ship between learning time, memorization, and cumulative sample loss. Figures 3 to 8 illustrate these relationships, showing an almost linear trend that suggests the bounds from the theorems are likely tight. The slight non-linearity between learning time and memorization likely arises from the theoretical assumption of uniformly bounded loss, while the cross-entropy loss used in practice does not have a uniform bound across all subpopulations (see Ravikumar et al. (2024a) for a related discussion).

### 6.2. Similarity with Memorization

**Experiment.** In this section, we examine how well our proposed proxy CSL correlates with the memorization score defined by Feldman & Zhang (2020). We also compare our proxy with those proposed by Garg et al. (2024), loss sensitivity (Arpit et al., 2017), forgetting frequency (Toneva et al., 2019), and final sample loss. This experiment trains an Inception model on CIFAR-100 and a ResNet50 model on ImageNet, using the same architectures as Feldman & Zhang (2020). For each dataset, we compute the memorization proxies and measure their cosine similarity with the memorization scores publicly made available by Feldman & Zhang (2020). This setup mirrors the approach used in Garg et al. (2024). Additionally, to capture another aspect of memorization, we plot the $L_2$ adversarial distance against both the memorization score and CSL for the CIFAR-100 dataset. Please see Appendix B.1 for more setup details.

**Results.** The results in Table 1 compare the cosine similarity between the proxies and two sets of examples: all exam-

ples and the Top-K memorized examples. "Top 5K" refers to the 5,000 most memorized examples, identified using Feldman & Zhang (2020), for which the cosine similarity is reported. For ImageNet, CSL proves to be the superior proxy. On top 5k of CIFAR-100, loss sensitivity has a slight edge over CSL. However, across all examples, CSL significantly outperforms other methods. Figure 9 plots the $L_2$ adversarial distance against both the memorization score and CSL for the CIFAR-100 dataset. Both exhibit similar trends: as memorization and CSL decrease, larger adversarial perturbations (i.e., greater distances) are required to flip the network's prediction.

**Takeaways.** CSL serves as an effective proxy for memorization, as demonstrated on both the CIFAR-100 and ImageNet datasets. Results across architectures (see Appendix B.2) confirm that these findings are consistent across architectures. CSL is computationally efficient, it is approximately 14× faster than curvature and 4 orders of magnitude faster than stability-based memorization. A breakdown of compute costs is provided in Appendix B.3. Additionally, CSL also captures the bimodal property (Lukasik et al., 2023) of memorization, as shown in Figures 12 and 13.

## 7. Theory to Practice

### 7.1. Mislabeled Detection

**Experiment.** In this section, we leverage insights from our theoretical framework to develop a practical method for detecting mislabeled examples in training datasets. We

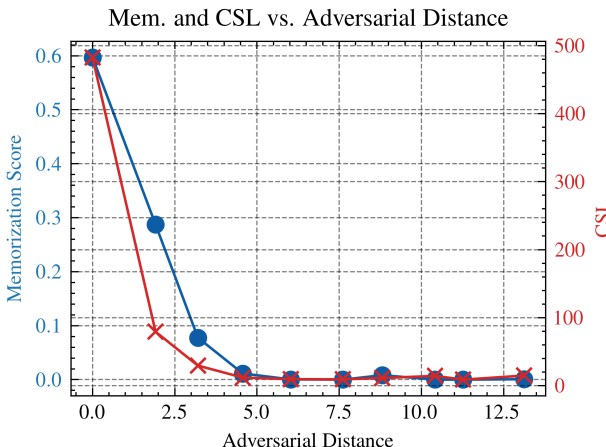

*Figure 9.* $L_2$ Adversarial distance vs Mem. score (Feldman & Zhang, 2020) and CSL on CIFAR-100. Clearly, memorization and CSL increases for less robust samples (i.e. lower adv distance $\|x - x_p\|_2$). Thus CSL captures similar properties as memorization

*Table 1.* CSL correlation and similarity with memorization compared to other methods across CIFAR-100 and ImageNet datasets. CS denotes cosine similarity and PC denotes Pearson correlation.

| Dataset | Arch. | Subset | Method | CS | PC |
|---|---|---|---|---|---|
| CIFAR-100 | Inception | Top 5k | Final Sample Loss | 0.33 | 0.06 |
| | | | Curv | 0.87 | 0.16 |
| | | | Loss Sensitivity | **0.97** | 0.39 |
| | | | Forget Freq. | 0.96 | 0.29 |
| | | | **CSL (Ours)** | 0.93 | **0.40** |
| | | All | Final Sample Loss | 0.24 | 0.17 |
| | | | Curv | 0.69 | 0.49 |
| | | | Loss Sensitivity | 0.81 | 0.76 |
| | | | Forget Freq. | 0.76 | 0.59 |
| | | | **CSL (Ours)** | **0.87** | **0.79** |
| ImageNet | ResNet50 | Top 50k | Final Sample Loss | 0.78 | 0.12 |
| | | | Curv | 0.84 | 0.05 |
| | | | Loss Sensitivity | 0.79 | 0.04 |
| | | | Forget Freq. | 0.68 | 0.15 |
| | | | **CSL (Ours)** | **0.94** | **0.21** |
| | | All | Final Sample Loss | 0.64 | 0.50 |
| | | | Curv | 0.62 | 0.33 |
| | | | Loss Sensitivity | 0.49 | 0.17 |
| | | | Forget Freq. | 0.49 | 0.04 |
| | | | **CSL (Ours)** | **0.79** | **0.64** |

evaluate the effectiveness of our approach by comparing it to several state-of-the-art methods for label error detection. The experiments are conducted on CIFAR-10 and CIFAR-100, where varying levels of symmetric label noise are introduced. Specifically, labels are randomly flipped to another class, uniformly across all classes, excluding the true label. To assess performance, we employ the Area

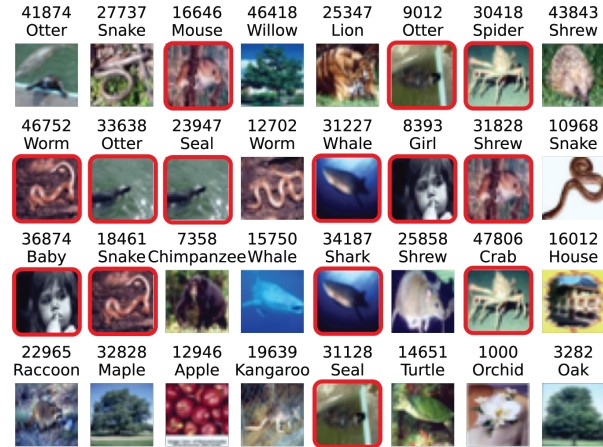

*Figure 10.* 32 highest scores for CSL on clean CIFAR-100 reveal conflicting labels, such as Crab and Spider outlined in red.

Under the Receiver Operating Characteristic (AUROC) metric, which measures the ability of each method to correctly identify mislabeled examples under different noise conditions. Additional details of the experiments and the baseline techniques are available in Appendix B.1 and B.5.

**Results.** The results are presented in Table 2, showcasing the performance of each method on CIFAR-10 and CIFAR-100 at symmetric label noise levels of 1%, 2%, 5%, and 10%. The term Thr. LT (Threshold based Learning Time) refers to the first epoch at which a sample is correctly classified (Jiang et al., 2021; Maini et al., 2022), reflecting the epoch at which the model learns a particular sample.

**Takeaways.** The proposed CSL proxy consistently outperforms other methods in detecting mislabeled examples across various label noise levels ranging from 1% to 10%.

**Compute Cost.** CSL incurs no additional computational overhead, unlike Confidence Learning (CL) (Northcutt et al., 2021), which requires training multiple models (3-folds in this case), significantly increasing cost. SSFT (Maini et al., 2022) approximately doubles training cost by training on two subsets, while input loss curvature demands 14 × more compute than CSL. Given that CSL and Thr. LT exhibit comparable computational costs, the superior performance and efficiency of CSL make it a highly compelling choice.

### 7.2. Duplicate Detection

**Experiment.** In this section, we apply the proposed memorization proxy to identify duplicate examples in the dataset. We conduct two types of analysis: first, a qualitative analysis of duplicate detection on the unmodified CIFAR-100 dataset; second, a quantitative experiment where we intentionally introduce duplicates (250 duplicates) into the dataset and use our proxy to identify them. We use a ResNet18 (He et al., 2016) model for this experiment, and the performance of our method is evaluated against other

*Table 2.* Evaluating the performance of mislabeled detection of the proposed framework against existing methods on CIFAR-10 and CIFAR-100 datasets under various levels of label noise.

| Dataset | Method | 1% Noise | 2% Noise | 5% Noise | 10% Noise |
|---------|--------|----------|----------|----------|-----------|
| CIFAR-10 | Thr. Learning Time (LT) | $0.4951 \pm 0.0248$ | $0.4954 \pm 0.0044$ | $0.4911 \pm 0.0071$ | $0.4948 \pm 0.0057$ |
| | In Conf. (Carlini et al., 2019a) | $0.8781 \pm 0.0177$ | $0.8072 \pm 0.0130$ | $0.7254 \pm 0.0214$ | $0.6528 \pm 0.0042$ |
| | CL (Northcutt et al., 2021) | $0.8651 \pm 0.0127$ | $0.8905 \pm 0.0115$ | $0.8874 \pm 0.0019$ | $0.8551 \pm 0.0030$ |
| | SSFT (Maini et al., 2022) | $0.9626 \pm 0.0018$ | $0.9551 \pm 0.0020$ | $0.9498 \pm 0.0042$ | $0.9360 \pm 0.0020$ |
| | Curv. (Garg et al., 2024) | $0.9715 \pm 0.0045$ | $0.9776 \pm 0.0033$ | $0.9800 \pm 0.0003$ | $0.9819 \pm 0.0006$ |
| | **CSL (Ours)** | $\mathbf{0.9845 \pm 0.0026}$ | $\mathbf{0.9864 \pm 0.0004}$ | $\mathbf{0.9870 \pm 0.0003}$ | $\mathbf{0.9869 \pm 0.0005}$ |
| CIFAR-100 | Thr. Learning Time (LT) | $0.5256 \pm 0.0012$ | $0.5227 \pm 0.0100$ | $0.5161 \pm 0.0051$ | $0.5203 \pm 0.0029$ |
| | In Conf. (Carlini et al., 2019a) | $0.7258 \pm 0.0102$ | $0.7236 \pm 0.0047$ | $0.7069 \pm 0.0069$ | $0.6884 \pm 0.0053$ |
| | CL (Northcutt et al., 2021) | $0.8723 \pm 0.0208$ | $0.8838 \pm 0.0006$ | $0.8733 \pm 0.0010$ | $0.8536 \pm 0.0006$ |
| | SSFT (Maini et al., 2022) | $0.8915 \pm 0.0045$ | $0.8893 \pm 0.0013$ | $0.8784 \pm 0.0030$ | $0.8664 \pm 0.0024$ |
| | Curv. (Garg et al., 2024) | $0.9856 \pm 0.0009$ | $0.9865 \pm 0.0011$ | $0.9876 \pm 0.0021$ | $0.9892 \pm 0.0012$ |
| | **CSL (Ours)** | $\mathbf{0.9891 \pm 0.0003}$ | $\mathbf{0.9895 \pm 0.0002}$ | $\mathbf{0.9895 \pm 0.0001}$ | $\mathbf{0.9897 \pm 0.0001}$ |

*Table 3.* Result of duplicate detection using the proposed methods and other baselines on CIFAR-10 and CIFAR-100 datasets.

| Method | CIFAR-10 | CIFAR-100 |
|--------|----------|-----------|
| Thr. LT | $0.7029 \pm 0.0058$ | $0.7419 \pm 0.0059$ |
| In Conf. | $0.9237 \pm 0.0114$ | $0.8623 \pm 0.0131$ |
| CL | $0.5533 \pm 0.0031$ | $0.5873 \pm 0.0090$ |
| SSFT | $0.8490 \pm 0.0034$ | $0.7938 \pm 0.0045$ |
| Curv. | $0.9536 \pm 0.0030$ | $0.9639 \pm 0.0030$ |
| **CSL (Ours)** | $\mathbf{0.9821 \pm 0.0006}$ | $\mathbf{0.9886 \pm 0.0008}$ |

techniques using AUROC (see Appendix B.1 for details).

**Results.** The qualitative analysis results are presented in Figure 10, which demonstrates the detection of duplicates in the clean CIFAR-100 dataset. The quantitative experimental results are provided in Table 3, where we report the AUROC scores for our method compared to other techniques.

**Takeaways.** As shown in Figure 10, CSL effectively identifies the majority of duplicates in the unmodified CIFAR-100 dataset. This is further validated in Table 3, where we evaluate the performance of the method after intentionally introducing duplicates. Here, we observe that CSL consistently achieves the best performance in identifying duplicates across both the CIFAR-10 and CIFAR-100 datasets. *To improve reproducibility, we have provided the code for all the experiments at* `https://github.com/DeepakTatachar/CSL-Mem`.

## 8. Conclusion

This paper provides a comprehensive theoretical framework that connects our memorization proxy, CSL, to learning time and stability-based memorization. We formalize the notion of learning time and our results demonstrate that CSL is not only highly effective in capturing memorization behavior but also computationally efficient, being four orders of magnitude faster than existing stability-based metrics.

We validate our framework through extensive experiments and show its practical applications in identifying mislabeled examples, and duplicates in datasets. The proposed proxy achieves state-of-the-art performance in identifying duplicate and mislabeled examples. By offering efficient tools to understand memorization, our framework can lead to more interpretable models across various machine learning tasks.

## Acknowledgements

This research was supported in part by the Center for the Co-Design of Cognitive Systems (COCOSYS), a DARPA-sponsored JUMP center, the Semiconductor Research Corporation (SRC), the National Science Foundation (NSF), and Collins Aerospace. The authors gratefully acknowledge the valuable feedback provided by Jimmy Gammell on earlier drafts of this manuscript.

## Impact Statement

The research presented in this paper significantly advances our understanding of memorization in deep learning models by introducing Cumulative Sample Loss (CSL) as a novel proxy for quantifying memorization. This work establishes a strong theoretical foundation connecting CSL to sample loss and stability-based memorization, demonstrating that CSL bounds stability-based memorization and learning time. By offering a proxy that is four orders of magnitude more computationally efficient than stability-based methods and incurs zero additional training overhead, this paper provides a practical and scalable tool for understanding and addressing memorization in deep models. This research not only deepens the theoretical comprehension of learning dynamics in deep models but also delivers actionable insights for improving training efficiency, robustness, and model quality. It offers transformative potential for fields reliant on deep learning, particularly in applications where data quality and computational efficiency are paramount.

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

# Appendix

## A. Stochastic Gradient Descent (SGD)

In SGD, model parameters $\vec{w}_t$ at iteration $t$ are updated using the gradient of the loss function computed with a mini-batch or a single random sample $\vec{z}_i$. The update rule is

$$\vec{w}_{t+1} = \vec{w}_t - \eta_t \tilde{\nabla}_{w_t} \ell(\vec{w}_t, \vec{z}_i) \tag{15}$$

Here, $\eta_t$ is the learning rate, and $\tilde{\nabla}_{w_t} \ell(\vec{w}_t, \vec{z}_i)$ is an unbiased stochastic gradient estimator.

**Bounded Gradient Variance.** Let $\nabla_{w_t} \ell(\mathbf{w}_t)$ denote the true gradient, and let $\widetilde{\nabla}_{w_t} \ell(\mathbf{w}_t)$ be an unbiased estimator of this gradient. The estimator is said to have variance bounded by $\Gamma_v^2$ if

$$\mathbb{E}\left[ \| \widetilde{\nabla}_{w_t} \ell(\vec{w}_t) - \nabla_{w_t} \ell(\vec{w}_t) \|_2^2 \right] \leq \Gamma_v^2 \tag{16}$$

## B. Experimental Details

### B.1. Setup

**Datasets.** We use CIFAR-10, CIFAR-100 (Krizhevsky et al., 2009) and ImageNet (Russakovsky et al., 2015) datasets. For experiments that use memorization scores, we use the pre-computed stability-based memorization scores from Feldman & Zhang (2020) which have been made publicly available by the authors.

**Architectures.** For all of experiments we train ResNet18 (He et al., 2016) models from scratch, expect for the cross architecture results in Table 4, where we train VGG16, MobileNetV2 and Inception (small inception as used by Feldman & Zhang (2020)). All the architectures used the same training recipe as described below.

**Training.** When training models on CIFAR-10 and CIFAR-100 the initial learning rate was set to 0.1 and trained for 200 epochs. The learning rate is decreased by 10 at epochs 120 and 180. When training on CIFAR-10 and CIFAR-100 datasets the batch size is set to 128. We use stochastic gradient descent for training with momentum set to 0.9 and weight decay set to 1e-4. For both CIFAR-10 and CIFAR-100 datasets, we used the following sequence of data augmentations for training: resize ($32 \times 32$), random crop, and random horizontal flip, this is followed by normalization. For ImageNet we trained a ResNet18 for 200 epochs with the same setting except the resize random crop was set to $224 \times 224$.

**Testing.** During testing no augmentations were used, i.e. we used resize followed by normalization. To improve reproducibility, we have provided the code in the supplementary material.

### B.2. Similarity with Memorization Scores across Architectures

**Experiment.** In this section, we present the results of measuring the cosine similarity between the proposed memorization proxy (CSL) and the memorization score from Feldman & Zhang (2020) across different architectures. Specifically, we tested VGG (Simonyan & Zisserman, 2014), MobileNetV2 (Sandler et al., 2018), and Inception (Szegedy et al., 2016).

**Results.** The results are shown in Table 4, which reports the cosine similarity between the CSL proxy and the memorization score for the three architectures on the CIFAR-100 dataset. These models were trained using the methodology described in Section B.1.

**Takeaways.** The results indicate that the top 5K i.e., the similarity between the metrics and the top 5000 most memorized samples according to Feldman & Zhang (2020) is highly consistent across architectures, and the overall match across the dataset is also quite high for CSL. However, two key observations are worth noting: (1) VGG16 shows a lower correlation on the 'All' category, and (2) Results on CSL are similar to the findings for ResNet18 in the main paper.

### B.3. Compute Cost Analysis

In this section, we provide a detailed analysis of the computational cost of the proposed proxies in comparison to other techniques. We assume the cost of one forward pass through a neural network is $F$, and consequently, the cost of a backpropagation step is $2F$, making the total cost for one forward-backward pass $3F$. Using previously defined notation, let $m$ represent the dataset size and $T$ the total number of training epochs.

| Architecture | Samples | CSL |
|---|---|---|
| VGG16 | Top 5K | 1.00 |
| | All | 0.61 |
| MobileNetV2 | Top 5K | 0.95 |
| | All | 0.73 |
| Inception | Top 5K | 0.97 |
| | All | 0.70 |

*Table 4.* Cosine similarity between stability-based memorization score with CSL for different architectures on CIFAR-100 for Top 5K and all samples.

**Stability-Based Memorization.** Feldman & Zhang (2020) trained between 2,000 and 10,000 models to compute the stability-based memorization score. Thus, the total computational cost is $10,000 \cdot 3F \cdot T \cdot m$.

**Cumulative Sample Curvature.** Garg et al. (2024) trained a single model and proposed using sample curvature averaged over training to estimate memorization. Hutchinson's trace estimator was employed to calculate curvature, which requires 2 forward passes and 1 backward pass, repeated $n$ times over the entire dataset for each epoch. While their results show that $n$ ranges from 2 to 10, we use $n = 2$ to provide the computational advantage in their favor, even though $n = 10$ produces better results. Thus, the total cost consists of the training and curvature computation.

$$\begin{aligned} \text{Cost} &= 3F \cdot T \cdot m + 4F \cdot T \cdot m \cdot n \\ &= 3F \cdot T \cdot m + 4F \cdot T \cdot m \cdot 2 \\ &= 11F \cdot T \cdot m \end{aligned}$$

If $n = 10$ is used for optimal results, as reported in Tables 2 and 3, the total computational cost becomes $43F \cdot T \cdot m$.

**CSL(Ours).** CSL can be obtained without additional computational cost during training. Therefore, the only cost is that of the training process, which is $3F \cdot T \cdot m$. The computational cost comparison is summarized in Table 5.

| Method | Absolute Cost | Relative Cost |
|---|---|---|
| Stability-Based (Feldman & Zhang, 2020) | $6000FTm - 30,000FTm$ | $2,000 \times - 30,000 \times$ |
| Cumulative Sample Curvature (Garg et al., 2024) | $11FTm - 43FTm$ | $3.6 \times - 14.33 \times$ |
| **CSL (Ours)** | $3FTm$ | $1 \times$ |

*Table 5.* Summary of the compute cost of the proposed metric compared to existing methods.

### B.4. Additional Details on Validating Theory Experiments

For the experiments described in Section 6.1, we provide additional details regarding the methodology. To generate the graphs in Figures 3, 4 and 5 we collected all relevant proxies for each sample in the dataset and grouped them into bins based on the x-axis metric (learning time or memorization score) in each figure. For instance, in Figure 4 samples were binned based on their learning time. Similarly, in Figure 5, we binned samples based on their memorization scores as defined in (Feldman & Zhang, 2020) and in Figure 3, the samples were again binned by learning time.

Since the theoretical framework is developed in terms of expected learning time, memorization, and the CSL paradigm, the standard deviation is not directly relevant to the interpretation of the results. Nevertheless, for the sake of completeness, we have visualized the standard deviation for each bin in Figure 11, even though it does not significantly contribute to the theoretical insights derived from expected scores.

**Learning Time Threshold $\tau$:** For this and all experiments where learning time is used, we follow the learning time

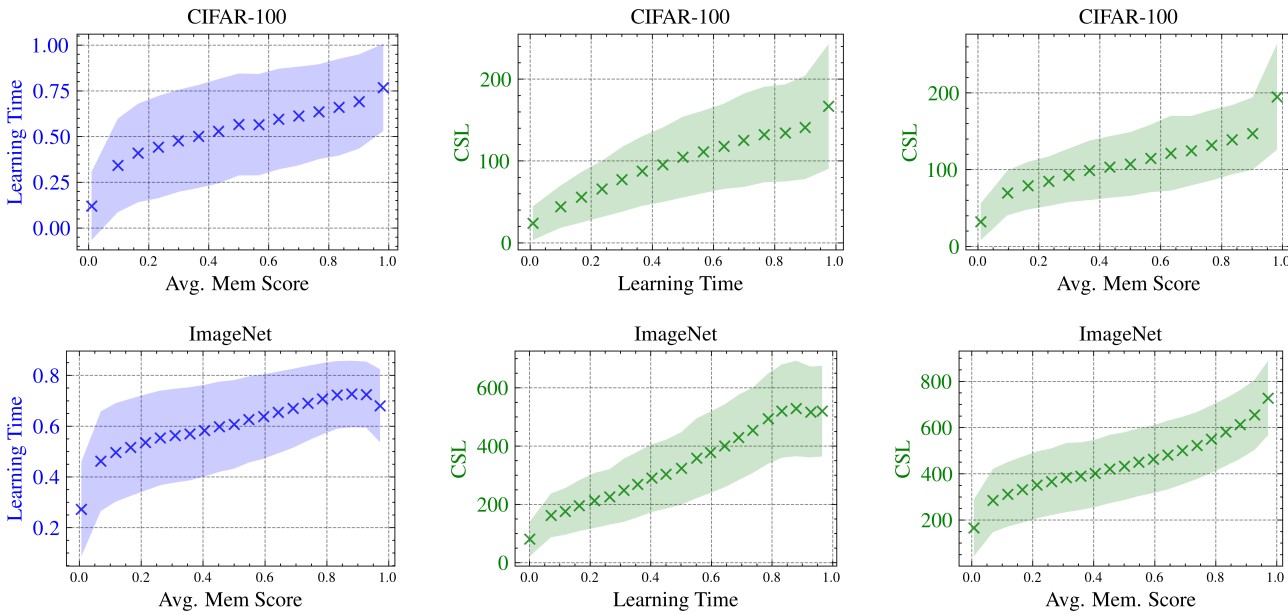

*Figure 11.* Validating theory results, linking learning time, memorization and CSL with error bars plotted as shaded area for CIFAR-100 and ImageNet.

definition (Eq. 7) and set the threshold $\tau$ to the average of the per iteration sample gradient, i.e. for example for ImageNet

$$\tau_{\text{ImageNet}} = E_{T \in \{0, \cdots T_{\max}\}, x_i \in \text{ImageNet}} \left[ \frac{1}{T} \sum_{t=0}^{T-1} \|\nabla_{x_i} \ell(\vec{w}_t)\|_2^2 \right] \tag{17}$$

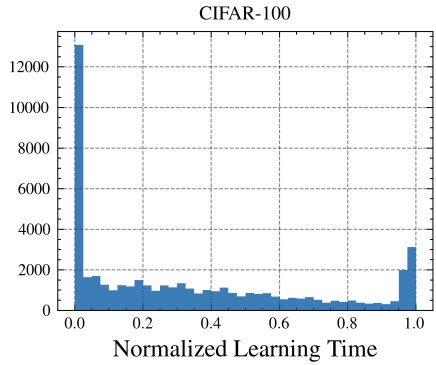

*Figure 12.* Histogram of learning times on CIFAR-100 dataset. Learning time captures the bimodal property of memorization score.

*Figure 13.* Histogram of learning times on the ImageNet dataset. Learning time captures the bimodal property of memorization score.

## B.5. Additional Details on Mislabelled Detection Experiments

In this section, we provide additional details regarding the setup for mislabel detection experiments. For all experiments, we trained the models using the training procedure outlined in Appendix B.1.

**In Confidence.** In-confidence (Carlini et al., 2019a) is calculated as 1 - "the predicted probability" of the true class.

**Confident Learning.** For the implementation of confident learning (Northcutt et al., 2021), we utilized the cleanlab library, which is available at `https://github.com/cleanlab/cleanlab`. We applied 3-fold cross-validation to compute out-of-sample probability scores for the samples. These probability scores were then input into the cleanlab implementation to generate the results reported.

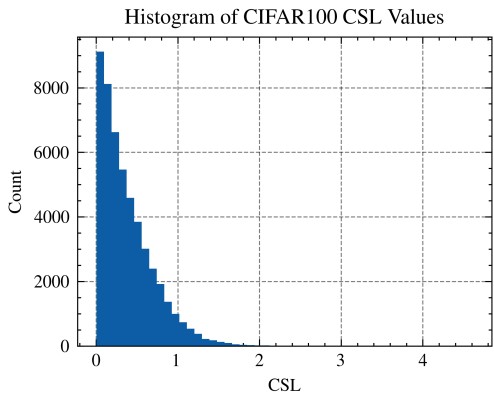

*Figure 14.* CSL Histogram on CIFAR100, captures long tail.

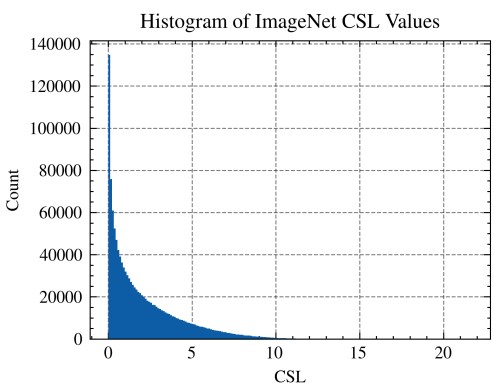

*Figure 15.* CSL Histogram on ImageNet, captures long tail.

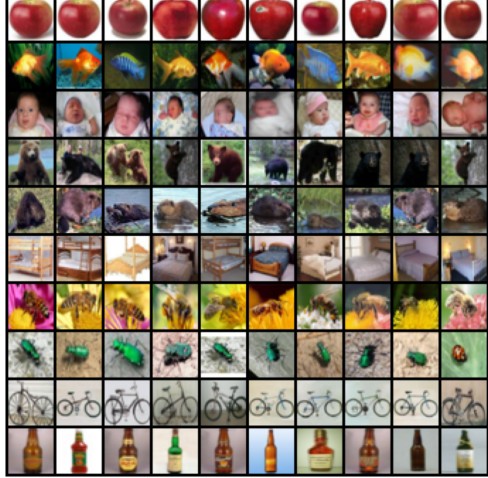

*Figure 16.* Lowest CSL Scores on CIFAR-100 captures easy-to-learn, Typical (likely generalized) examples. Visualized for the first 10 classes.

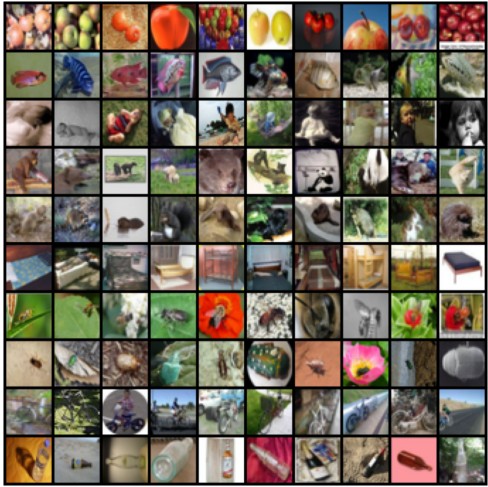

*Figure 17.* Highest CSL Scores on CIFAR-100 capture hard-to-learn, Atypical (likely memorized) examples. Visualized for the first 10 classes.

**SSFT.** Second Split Forgetting Time (SSFT) (Maini et al., 2022) is measured using two subsets, Set 1 and Set 2. A model is first trained on Set 1 and subsequently fine-tuned on Set 2, during which we measure how quickly a sample from Set 1 is misclassified or "forgotten". This process is repeated for both subsets to cover the entire dataset. Specifically, after training on Set 1 and measuring the forgetting time for samples in Set 1 during fine-tuning on Set 2, the model is then trained on Set 2, and the forgetting time for Set 2 is measured during fine-tuning on Set 1.

**Thr. LT. (Threshold based Learning Time).** Consider a sample's correct predictions during training over 12 epochs in this case. Let the correct predictions be $[0, 0, 0, 1, 1, 0, 1, 1, 0, 1, 1, 1]$. Then Thr. LT would be calculated as epoch 3 (beginning at 0). This is normalized so that it's between 0 - 1.

**Curvature.** To calculate the curvature of a sample, we used the technique described in Garg et al. (2024). The hyperparameters were set to $n = 10$ and $h = 0.001$, following the same configuration as outlined by Garg et al. (2024).

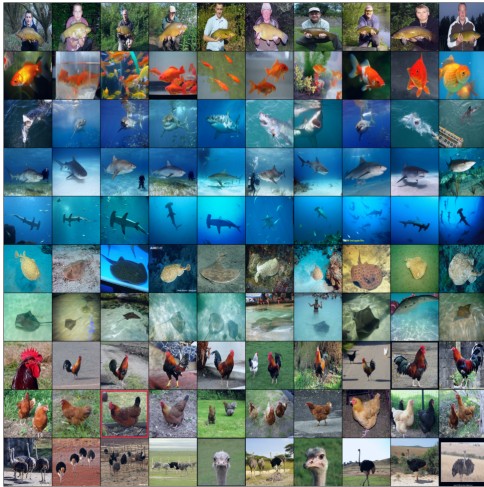

*Figure 18.* Lowest CSL Scores on ImageNet captures easy-to-learn, Typical (likely generalized) examples. Visualized for the first 10 classes.

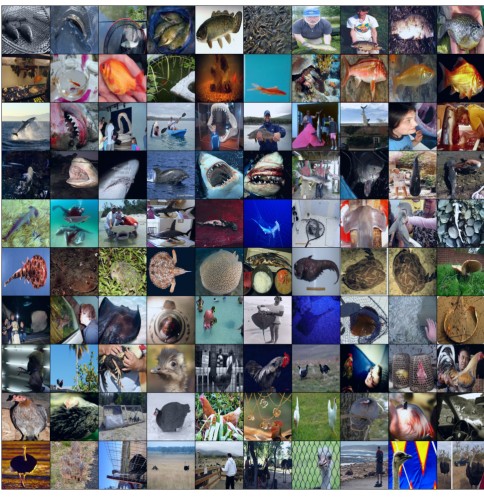

*Figure 19.* Highest CSL Scores on ImageNet capture hard-to-learn, Atypical (likely memorized) examples. Visualized for the first 10 classes.

## C. Proofs

### C.1. Proof of Lemma 5.1

**Lemma 5.1** *For any neural network and a given mini-batch of inputs $Z_b = (X_b, Y_b)$, the Frobenius norm of the gradient of the loss $\ell$ with respect to the input is bounded by the norm of the gradient with respect to the network's weights $\vec{w}_t$. Specifically:*

$$\|\nabla_{Z_b} \ell(\vec{w}_t, Z_b)\|_F \leq k_g \|\nabla_{w_t} \ell(\vec{w}_t, Z_b)\|_F \tag{18}$$

*where $k_g = \dfrac{\|W_t^{(1)}\|_F \, \|(X_b^\top)^+\|_F}{s_P}$ and $s_P$ denotes the smallest singular value of $P = X_b^\top (X_b^\top)^+$, where $^+$ denotes pseudo-inverse.*

**Proof**

Let the row vector $\vec{w}_t = [\vec{w}_t^{(1)}, \vec{w}_t^{(2)}, \cdots, \vec{w}_t^{(q)}]$ denote the weights of a $q$-layered neural network at iteration $t$, where

$$\vec{w}_t^{(k)} = \begin{bmatrix} w_{t,1,1}^{(k)} & w_{t,1,2}^{(k)} & \cdots & w_{t,d_k,d_{k-1}}^{(k)} \end{bmatrix} \in \mathbb{R}^{d_k \cdot d_{k-1}} \tag{19}$$

is a flattened row vector representing the weights of the $k^{\text{th}}$ layer at iteration $t$, with input dimension $d_{k-1}$ and output dimension $d_k$. This vector can be reshaped into a matrix $W_t^{(k)} \in \mathbb{R}^{d_k \times d_{k-1}}$ by rearranging its elements into the specified dimensions $d_k \times d_{k-1}$ while maintaining the order of elements. The reshaping operation is formally defined as:

$$W_t^{(k)} = \text{Reshape}(\vec{w}_t^{(k)}) = \begin{bmatrix} w_{t,1,1}^{(k)} & w_{t,1,2}^{(k)} & \cdots & w_{t,1,d_{k-1}}^{(k)} \\ w_{t,2,1}^{(k)} & w_{t,2,2}^{(k)} & \cdots & w_{t,2,d_{k-1}}^{(k)} \\ \vdots & \vdots & \ddots & \vdots \\ w_{t,d_k,1}^{(k)} & w_{t,d_k,2}^{(k)} & \cdots & w_{t,d_k,d_{k-1}}^{(k)} \end{bmatrix} \tag{20}$$

For our initial analysis, we consider a two-layer neural network given by:

$$IR^{(1)} = \sigma(W_t^{(1)} X_b) \tag{21}$$

$$\hat{Y} = W_t^{(2)} IR^{(1)}, \tag{22}$$

where $W_t^{(1)} \in \mathbb{R}^{d_1 \times d_0}$ and $W_t^{(2)} \in \mathbb{R}^{d_2 \times d_1}$ are matrices constructed from $\vec{w}_t^{(1)}$ and $\vec{w}_t^{(2)}$, respectively. Here, $X_b \in \mathbb{R}^{d_0 \times b}$ represents the input batch, $IR^{(1)} \in \mathbb{R}^{d_1 \times b}$ denotes the intermediate representation after applying the activation function

$\sigma(\cdot)$, and $\hat{Y} \in \mathbb{R}^{d_2 \times m}$ represents the predicted output. In our case $d_0 = n$ and $d_2 = 1$. Let the labels for the batch $X_b$ be represented as $Y_b = [y_1, \cdots y_b]$, the we can use chain rule to write the gradients with respect to input and weight parameters as below

$$
\begin{aligned}
\nabla_{X_b}\ell(\vec{w}_t, Z_b) &= \nabla_{\hat{Y}}\ell(\vec{w}_t, Z_b)\, \nabla_{IR^{(1)}}\hat{Y}\, \nabla_{X_b}IR^{(1)} \\
&= (W_t^{(1)})^T\, (W_t^{(2)})^T\nabla_{\hat{Y}}\ell(\vec{w}_t, Z_b) \odot \sigma'(IR^{(1)}) \\
&= (W_t^{(1)})^T\, (W_t^{(2)})^T\nabla_{\hat{Y}}\ell(\vec{w}_t, Z_b) \odot \sigma'(W_t^{(1)}X_b)
\end{aligned}
\tag{23}
$$

Similarly for the gradient w.r.t to $W_t^{(1)}$ we have

$$
\begin{aligned}
\nabla_{W_t^{(1)}}\ell(\vec{w}_t, Z_b) &= \nabla_{\hat{Y}}\ell(\vec{w}_t, Z_b)\, \nabla_{IR^{(1)}}\hat{Y}\, \nabla_{W_t^{(1)}}IR^{(1)} \\
&= (W_t^{(2)})^T\nabla_{\hat{Y}}\ell(\vec{w}_t, Z_b) \odot \sigma'(IR^{(1)})\, X_b^T \\
&= (W_t^{(2)})^T\nabla_{\hat{Y}}\ell(\vec{w}_t, Z_b) \odot \sigma'(W_t^{(1)}X_b)\, X_b^T
\end{aligned}
\tag{24}
$$

From Equations 23 and 24 we have:

$$
(W_t^{(1)})^T\nabla_{W_t^{(1)}}\ell(\vec{w}_t, Z_b) = \nabla_{X_b}\ell(\vec{w}_t, Z_b)X_b^T
\tag{25}
$$

$$
(W_t^{(1)})^T\nabla_{W_t^{(1)}}\ell(\vec{w}_t, Z_b)(X_b^T)^+ = \nabla_{X_b}\ell(\vec{w}_t, Z_b)X_b^T(X_b^T)^+
$$

$$
\|(W_t^{(1)})^T\nabla_{W_t^{(1)}}\ell(\vec{w}_t, Z_b)(X_b^T)^+\|_F = \|\nabla_{X_b}\ell(\vec{w}_t, Z_b)X_b^T(X_b^T)^+\|_F
\tag{26}
$$

Now note that Equations 25 and 26 hold for any deep neural network. Next if we let $s_P$ denote the smallest singular value of $P = X_b^T(X_b^T)^+$

$$
\|(W_t^{(1)})^T\|_F\, \|\nabla_{W_t^{(1)}}\ell(\vec{w}_t, Z_b)\|_F\, \|(X_b^T)^+\|_F \geq \|\nabla_{X_b}\ell(\vec{w}_t, Z_b)X_b^T(X_b^T)^+\|_F
$$

$$
\|(W_t^{(1)})^T\|_F\, \|\nabla_{W_t^{(1)}}\ell(\vec{w}_t, Z_b)\|_F\, \|(X_b^T)^+\|_F \geq \|\nabla_{X_b}\ell(\vec{w}_t, Z_b)X_b^T(X_b^T)^+\|_F \geq s_P\|\nabla_{X_b}\ell(\vec{w}_t, Z_b)\|_F
$$

Thus we have

$$
\|\nabla_{X_b}\ell(\vec{w}_t, Z_b)\|_F \leq \frac{\|(W_t^{(1)})^T\|_F\, \|(X_b^T)^+\|_F}{s_P}\|\nabla_{W_t^{(1)}}\ell(\vec{w}_t, Z_b)\|_F
$$

$$
\|\nabla_{X_b}\ell(\vec{w}_t, Z_b)\|_F \leq k_g\|\nabla_{W_t^{(1)}}\ell(\vec{w}_t, Z_b)\|_F
$$

Now observe that $\|\nabla_{W_t^{(1)}}\ell(\vec{w}_t, Z_b)\|_F \leq \|\nabla_{w_t}\ell(\vec{w}_t, Z_b)\|_F$ where $W_t$ is the weights of the entire network (see proof in section C.1.1). Intuitively, the gradient with respect to a single layer is just a portion of the overall gradient vector, and removing coordinates (the other layers' gradients) cannot increase the Frobenius (or $\ell_2$) norm.

$$
\|\nabla_{X_b}\ell(\vec{w}_t, Z_b)\|_F \leq \frac{\|(W_t^{(1)})^T)\|_F\, \|(X_b^T)^+\|_F}{s_P}\|\nabla_{W_t}\ell(\vec{w}_t, Z_b)\|_F
$$

$$
\|\nabla_{X_b}\ell(\vec{w}_t, Z_b)\|_F \leq k_g\|\nabla_{w_t}\ell(\vec{w}_t, Z_b)\|_F \quad \blacksquare
$$

### C.1.1. WEIGHT NORM RELATION PROOF

We aim to prove that

$$
\|\nabla_{W_t^{(1)}}\ell(\vec{w}_t, Z_b)\|_F \leq \|\nabla_{w_t}\ell(\vec{w}_t, Z_b)\|_F,
\tag{27}
$$

We consider a $p$-layer neural network whose entire weight vector at iteration $t$ is

$$
w_t = \left[\vec{w}_t^{(1)}, \vec{w}_t^{(2)}, \ldots, \vec{w}_t^{(p)}\right],
$$

where $\vec{w}_t^{(k)}$ is the flattened vector of parameters in the $k$th layer.

The gradient of the loss $\nabla_{w_t} \ell(\vec{w}_t, Z_b)$ can be viewed as the concatenation of the gradients with respect to each layer:

$$\nabla_{w_t} \ell(\vec{w}_t, Z_b) = \begin{bmatrix} \nabla_{w_t^{(1)}} \ell \\ \nabla_{w_t^{(2)}} \ell \\ \vdots \\ \nabla_{w_t^{(p)}} \ell \end{bmatrix}.$$

In particular, $\nabla_{W_t^{(1)}} \ell$ is just the top block (corresponding to the first layer) of the entire gradient vector.

Let us denote

$$g^{(1)} := \nabla_{w_t^{(1)}} \ell(\vec{w}_t, Z_b), \quad g^{(2)} := \nabla_{w_t^{(2)}} \ell(\vec{w}_t, Z_b), \quad \dots, \quad g^{(p)} := \nabla_{w_t^{(p)}} \ell(\vec{w}_t, Z_b).$$

Then the full gradient can be written as

$$G := \nabla_{w_t} \ell(\vec{w}_t, Z_b) = \begin{bmatrix} g^{(1)} \\ g^{(2)} \\ \vdots \\ g^{(p)} \end{bmatrix}$$

By the definition of the Frobenius norm (which coincides with the Euclidean $\ell_2$ norm on the flattened vector), we have

$$\begin{aligned} \|G\|_F^2 &= \left\|g^{(1)}\right\|_F^2 + \left\|g^{(2)}\right\|_F^2 + \cdots + \left\|g^{(p)}\right\|_F^2 \\ &\geq \left\|g^{(1)}\right\|_F^2. \end{aligned} \tag{28}$$

Taking the square root on both sides preserves the inequality:

$$\|G\|_F = \sqrt{\|g^{(1)}\|_F^2 + \cdots + \|g^{(p)}\|_F^2} \geq \sqrt{\|g^{(1)}\|_F^2} = \|g^{(1)}\|_F. \tag{29}$$

Since $g^{(1)} = \nabla_{W_t^{(1)}} \ell(\vec{w}_t, Z_b)$ and $G = \nabla_{w_t} \ell(\vec{w}_t, Z_b)$, we obtain

$$\|\nabla_{W_t^{(1)}} \ell(\vec{w}_t, Z_b)\|_F \leq \|\nabla_{w_t} \ell(\vec{w}_t, Z_b)\|_F \quad \blacksquare \tag{30}$$

Intuitively, the gradient with respect to a single layer is just a portion of the overall gradient vector, and removing coordinates (the other layers' gradients) cannot increase the Frobenius (or $\ell_2$) norm.

## D. Proofs

### D.1. Proof of Theorem 5.2

**Proof**

**Assumptions**:

- Assume that the loss function $\ell$ is $\rho$-Lipschitz.

- Assume that the gradient estimator $\widetilde{\nabla}_w \ell$ is unbiased i.e., $\mathbb{E}_t[\widetilde{\nabla}_w \ell(\vec{w}_t)] = \nabla_w \ell(\vec{w}_t)$

- Assume bounded gradient variance $\mathbb{E}\left[\|\widetilde{\nabla}_{w_t} \ell(\vec{w}_t) - \nabla_{w_t} \ell(\vec{w}_t)\|_2^2\right] \leq \Gamma_v^2$.

- Assume $\mathbb{E}_t\left[\langle \nabla_{w_t} \ell(\vec{w}_t), \delta_t \rangle\right] = 0$

**Proof**

Let $\vec{w}_t$ denote the vector of the weight parameters at the $t^{th}$ iteration. For simplicity, we slightly abuse the notation by using $\nabla_w = \nabla_{\vec{w}_t}$ throughout the paper for ease of reference. Using the $\rho$-Lipschitz assumption on the loss, for any vectors $\vec{w}_t$ and $\vec{w}_{t+1}$, the function $\ell$ satisfies the quadratic upper bound:

$$\ell(\vec{w}_{t+1}) \leq \ell(\vec{w}_t) + \langle \nabla_w \ell(\vec{w}_t), \vec{w}_{t+1} - \vec{w}_t \rangle + \frac{\rho}{2}\|\vec{w}_t - \vec{w}_{t+1}\|_2^2 \tag{31}$$

We know the SGD update equation is

$$\vec{w}_{t+1} = \vec{w}_t - \eta_t \widetilde{\nabla}_{w_t}\ell(\vec{w}_t) \tag{32}$$

Using the SGD update in Eq. 31 we get

$$\ell(\vec{w}_{t+1}) \leq \ell(\vec{w}_t) - \langle \nabla_w\ell(\vec{w}_t), \eta_t\widetilde{\nabla}_{w_t}\ell(\vec{w}_t)\rangle + \frac{\rho\eta_t^2}{2}\|\widetilde{\nabla}_{w_t}\ell(\vec{w}_t)\|_2^2 \tag{33}$$

$$\ell(\vec{w}_{t+1}) \leq \ell(\vec{w}_t) - \eta_t\langle \nabla_w\ell(\vec{w}_t), \widetilde{\nabla}_{w_t}\ell(\vec{w}_t)\rangle + \frac{\rho\eta_t^2}{2}\|\widetilde{\nabla}_{w_t}\ell(\vec{w}_t)\|_2^2 \tag{34}$$

Following the trick from the SGD convergence proof by (Ghadimi & Lan, 2013) (for randomized stochastic gradient method), we have $\delta_k = \widetilde{\nabla}_w\ell(\vec{w}_t) - \nabla_w\ell(\vec{w}_t)$

$$\ell(\vec{w}_{t+1}) \leq \ell(\vec{w}_t) - \eta_t\langle \nabla_w\ell(\vec{w}_t), \nabla_{w_t}\ell(\vec{w}_t)\rangle - \eta_t\langle \nabla_w\ell(\vec{w}_t), \delta_t\rangle + \frac{\rho\eta_t^2}{2}\|\widetilde{\nabla}_{w_t}\ell(\vec{w}_t)\|_2^2 \tag{35}$$

$$\ell(\vec{w}_{t+1}) \leq \ell(\vec{w}_t) - \eta_t\langle \nabla_w\ell(\vec{w}_t), \nabla_{w_t}\ell(\vec{w}_t)\rangle - \eta_t\langle \nabla_w\ell(\vec{w}_t), \delta_t\rangle + \frac{\rho\eta_t^2}{2}\|\widetilde{\nabla}_{w_t}\ell(\vec{w}_t)\|_2^2 \tag{36}$$

Taking conditional expectation at timestep $t$ we get

$$\mathbb{E}_t\left[\ell(\vec{w}_{t+1})\right] \leq \ell(\vec{w}_t) - \eta_t\|\nabla_{w_t}\ell(\vec{w}_t)\|_2^2 - 0 + \frac{\rho\eta_t^2}{2}\mathbb{E}_t\left[\|\widetilde{\nabla}_{w_t}\ell(\vec{w}_t)\|_2^2\right] \tag{37}$$

$$\mathbb{E}_t\left[\ell(\vec{w}_{t+1})\right] \leq \ell(\vec{w}_t) - \eta_t\|\nabla_{w_t}\ell(\vec{w}_t)\|_2^2 - 0 + \frac{\rho\eta_t^2}{2}\mathbb{E}_t\left[\|\nabla_{w_t}\ell(\vec{w}_t)\|_2^2 + 2\langle \nabla_{w_t}\ell(\vec{w}_t), \delta_t\rangle + \|\delta_t\|_2^2\right] \tag{38}$$

$$\mathbb{E}_t\left[\ell(\vec{w}_{t+1})\right] \leq \ell(\vec{w}_t) - \left(\eta_t - \frac{\rho\eta_t^2}{2}\right)\|\nabla_{w_t}\ell(\vec{w}_t)\|_2^2 + \frac{\rho\eta_t^2}{2}\mathbb{E}_t\left[2\langle \nabla_{w_t}\ell(\vec{w}_t), \delta_t\rangle + \|\delta_t\|_2^2\right] \tag{39}$$

Using the bounded variance assumption and $\mathbb{E}_t\left[\langle \nabla_{w_t}\ell(\vec{w}_t), \delta_t\rangle\right] = 0$, we have

$$\mathbb{E}_t\left[\ell(\vec{w}_{t+1})\right] \leq \mathbb{E}_t[\ell(\vec{w}_t)] - \left(\eta_t - \frac{\rho\eta_t^2}{2}\right)\mathbb{E}_t\left[\|\nabla_{w_t}\ell(\vec{w}_t)\|_2^2\right] + \frac{\rho\eta_t^2\Gamma_v^2}{2} \tag{40}$$

Rearranging and telescoping we get

$$\left(\eta_t - \frac{\rho\eta_t^2}{2}\right)\mathbb{E}_t\left[\|\nabla_{w_t}\ell(\vec{w}_t)\|_2^2\right] \leq \mathbb{E}_t[\ell(\vec{w}_t)] - \mathbb{E}_t\left[\ell(\vec{w}_{t+1})\right] + \frac{\rho\eta_t^2\Gamma_v^2}{2} \tag{41}$$

$$\sum_{t=0}^{T-1}\left(\eta_t - \frac{\rho\eta_t^2}{2}\right)\mathbb{E}_t\left[\|\nabla_{w_t}\ell(\vec{w}_t)\|_2^2\right] \leq \mathbb{E}_t\left[\ell(\vec{w}_0) - \ell(\vec{w}_T)\right] + \frac{\rho\Gamma_v^2}{2}\sum_{t=0}^{T-1}\eta_t^2 \tag{42}$$

Let's divide each side by

$$\sum_{t=0}^{T-1}\left(\eta_t - \frac{\rho\eta_t^2}{2}\right)$$

Now consider:

$$\frac{\sum_{t=0}^{T-1}\left(\eta_t - \frac{\rho\eta_t^2}{2}\right)\mathbb{E}_t\left[\|\nabla_{w_t}\ell(\vec{w}_t)\|_2^2\right]}{\sum_{t=0}^{T-1}\left(\eta_t - \frac{\rho\eta_t^2}{2}\right)}$$

This is nothing but a weighted sum i.e. expectation, since we hold the choice of the weights, we can chose them so that it is uniformly randomly distributed. Thus we can write the following below where $R$ is such a variable.

$$\mathbb{E}_R\left[\|\nabla_w\ell(\vec{w}_R)\|_2^2\right] \leq \frac{1}{\sum_{t=0}^{T-1}\left(\eta_t - \frac{\rho\eta_t^2}{2}\right)}\mathbb{E}_t\left[\ell(\vec{w}_0) - \ell(\vec{w}_T)\right]$$

$$+ \frac{\rho\Gamma_v^2}{2}\cdot\frac{\sum_{t=0}^{T-1}\eta_t^2}{\sum_{t=0}^{T-1}\left(\eta_t - \frac{\rho\eta_t^2}{2}\right)} \tag{43}$$

The above the is standard convergence result for SGD from Ghadimi & Lan (2013). Let us now consider Eq. 41.

$$\left(\eta_t - \frac{\rho\eta_t^2}{2}\right)\mathbb{E}_t\left[\|\nabla_{w_t}\ell(\vec{w}_t)\|_2^2\right] \leq \mathbb{E}_t[\ell(\vec{w}_t)] - \mathbb{E}_t\left[\ell(\vec{w}_{t+1})\right] + \frac{\rho\eta_t^2\Gamma_v^2}{2} \tag{44}$$

From Lemma 5.1 we know the input and weight gradient are related by $\kappa_g$, this varies as a function of $\vec{w}_t$. But there exists a constant $\kappa_g^{t_m}$ which satisfies Lemma 5.1 for all $t$. Trivially, $\kappa_g^{t_m} = \max_t \kappa_g^t$, since $\kappa_g^t > 0$. Let $\kappa_m = \max_t(\kappa_g^t)^2$. Now multiply both sides by $\kappa_m$, we get

$$\left(\eta_t - \frac{\rho\eta_t^2}{2}\right)\mathbb{E}_t\left[\|\nabla_{x_i}\ell(\vec{w}_t)\|_2^2\right] \leq \kappa_m\left(\mathbb{E}_t[\ell(\vec{w}_t)] - \mathbb{E}_t\left[\ell(\vec{w}_{t+1})\right]\right) + \frac{(\kappa_g^t\eta_t)^2\rho\Gamma_v^2}{2} \tag{45}$$

Summing over $t = 0$ to $t = T - 1$ we get

$$\sum_{t=0}^{T-1}\left(\eta_t - \frac{\rho\eta_t^2}{2}\right)\mathbb{E}_t\left[\|\nabla_{x_i}\ell(\vec{w}_t)\|_2^2\right] \leq \kappa_m\sum_{t=0}^{T-1}\left(\mathbb{E}_t[\ell(\vec{w}_t)] - \mathbb{E}_t\left[\ell(\vec{w}_{t+1})\right]\right) + \frac{\rho\Gamma_v^2}{2}\sum_{t=0}^{T-1}(\kappa_g^t\eta_t)^2 \tag{46}$$

$$\sum_{t=0}^{T-1}\left(\eta_t - \frac{\rho\eta_t^2}{2}\right)\mathbb{E}_t\left[\|\nabla_{x_i}\ell(\vec{w}_t)\|_2^2\right] \leq \kappa_m\left(\mathbb{E}_t[\ell(\vec{w}_0)] - \mathbb{E}_t\left[\ell(\vec{w}_T)\right] + \frac{\rho\Gamma_v^2}{2}\sum_{t=0}^{T-1}(\kappa_g^t\eta_t)^2 \tag{47}$$

Let us divide on each by $\sum_{t=0}^{T-1}\left(\eta_t - \frac{\rho\eta_t^2}{2}\right)$

$$\frac{\sum_{t=0}^{T-1}\left(\eta_t - \frac{\rho\eta_t^2}{2}\right)\mathbb{E}_t\left[\|\nabla_{x_i}\ell(\vec{w}_t)\|_2^2\right]}{\sum_{t=0}^{T-1}\left(\eta_t - \frac{\rho\eta_t^2}{2}\right)} \leq \frac{\kappa_m}{\sum_{t=0}^{T-1}\left(\eta_t - \frac{\rho\eta_t^2}{2}\right)}\left(\mathbb{E}_t[\ell(\vec{w}_0)] - \mathbb{E}_t\left[\ell(\vec{w}_T)\right] + \left(\frac{\rho\Gamma_v^2}{2}\right)\frac{\sum_{t=0}^{T-1}(\kappa_g^t\eta_t)^2}{\sum_{t=0}^{T-1}\left(\eta_t - \frac{\rho\eta_t^2}{2}\right)} \tag{48}$$

Using the weighted sum argument, we can write this as follows

$$\mathbb{E}_R[\|\nabla_{x_i}\ell(\vec{w}_R)\|_2^2] \leq \frac{\kappa_m}{\sum_{t=0}^{T-1}\left(\eta_t - \frac{\rho\eta_t^2}{2}\right)}\left(\mathbb{E}_t[\ell(\vec{w}_0)] - \mathbb{E}_t\left[\ell(\vec{w}_T)\right] + \left(\frac{\rho\Gamma_v^2}{2}\right)\frac{\sum_{t=0}^{T-1}(\kappa_g^t\eta_t)^2}{\sum_{t=0}^{T-1}\left(\eta_t - \frac{\rho\eta_t^2}{2}\right)} \tag{49}$$

Now let $\eta_t = \dfrac{1}{\sqrt{T}}$, consider the first term on the right hand side

$$\frac{\kappa_m}{\sum_{t=0}^{T-1}\left(\eta_t - \dfrac{\rho\eta_t^2}{2}\right)}\left(\mathbb{E}_t[\ell(\vec{w}_0)] - \mathbb{E}_t\left[\ell(\vec{w}_T)\right]\right) = O\left(\frac{1}{T/\sqrt{T}}\right) = O\left(\frac{1}{\sqrt{T}}\right) \tag{50}$$

Let us now consider the second term

$$\left(\frac{\rho\Gamma_v^2}{2}\right)\frac{\sum_{t=0}^{T-1}(\kappa_g^t\eta_t)^2}{\sum_{t=0}^{T-1}\left(\eta_t - \dfrac{\rho\eta_t^2}{2}\right)} = O\left(\frac{\frac{T}{T}}{\frac{T}{\sqrt{T}}}\right) = O\left(\frac{1}{\sqrt{T}}\right) \tag{51}$$

Thus we get the convergence result

$$\mathbb{E}_R\left[\|\nabla_{x_i}\ell(\vec{w}_R)]\|_2^2\right] = O\left(\frac{1}{\sqrt{T}}\right) \quad \blacksquare \tag{52}$$

where $R$ is the result iterate and $T$ is an iterate greater than $R$. Additionally, see Corollary 2.2 from Ghadimi & Lan (2013) where $\eta_t = \min\left(\dfrac{1}{\rho}, \dfrac{\widetilde{D}}{\Gamma_v\sqrt{T}}\right)$ for some $\widetilde{D} > 0$, and taking the tower expectation over the iterate $t$ (see Ghadimi & Lan (2013) for more details) we get the above convergence result.

### D.2. Learning Time and Memorization

**Theorem 5.3** *(Learning Time bounds Memorization) Under the assumptions of SGD convergence and L-bounded loss, there exists a $\kappa_T > 0$ such that the expected learning time for a sample $\vec{z}_i$ bounds expected memorization score as follows.*

$$\mathbb{E}_{z_i}\left[\operatorname{mem}(\vec{z}_i)\right] \leq \kappa_T\,\mathbb{E}_{z_i,p}\left[T_{z_i}\right] \tag{53}$$

**Proof of Theorem 5.3**

**Assumptions**:

- Assume that the loss function $\ell$ is $\rho$-Lipschitz.

- Assume that the gradient estimator $\widetilde{\nabla}_w\ell$ is unbiased i.e., $\mathbb{E}_t[\widetilde{\nabla}_w\ell(\vec{w}_t)] = \nabla_w\ell(\vec{w}_t)$

- Assume bounded gradient variance $\mathbb{E}\left[\|\widetilde{\nabla}_{w_t}\ell(\vec{w}_t) - \nabla_{w_t}\ell(\vec{w}_t)\|_2^2\right] \leq \Gamma_v^2$.

- Assume loss is bounded $0 \leq \ell \leq L$

- Assume $\beta$-stability as stated in Assumption 4.

**Proof**

Consider Equation 48:

$$\mathbb{E}_R\left[\|\nabla_{x_i}\ell(\vec{w}_R)]\right] \leq \frac{\kappa_m}{\sum_{t=0}^{T-1}\left(\eta_t - \dfrac{\rho\eta_t^2}{2}\right)}\left(\mathbb{E}_t[\ell(\vec{w}_0)] - \mathbb{E}_t\left[\ell(\vec{w}_T)\right]\right) + \left(\frac{\rho\Gamma_v^2}{2}\right)\frac{\sum_{t=0}^{T-1}(\kappa_g^t\eta_t)^2}{\sum_{t=0}^{T-1}\left(\eta_t - \dfrac{\rho\eta_t^2}{2}\right)} \tag{54}$$

Let

$$\eta = \eta_t - \frac{\rho\eta_t^2}{2} \tag{55}$$

$$\eta_s = \frac{\sum_{t=0}^{T-1}(\eta_t)^2}{\sum_{t=0}^{T-1}\left(\eta_t - \dfrac{\rho\eta_t^2}{2}\right)} \tag{56}$$

Using the above we have, $\eta_s$ is independent of $T$ and for the worst case bound we can set $\eta = \min_t \left( \eta_t - \frac{\rho \eta_t^2}{2} \right)$. Thus we can write the following

$$\mathbb{E}_R \left[ \| \nabla_{x_i} \ell(\vec{w}_R) \|_2^2 \right] \leq \frac{\kappa_m}{T\eta} \left( \mathbb{E}_t \left[ \ell(\vec{w}_0) - \ell(\vec{w}_T) \right] \right) + \frac{\kappa_m \eta_s \rho \Gamma^2}{2} \tag{57}$$

$$\leq \frac{\kappa_m}{T\eta} \left( \mathbb{E}_t \left[ \ell(\vec{w}_0) - \ell(\vec{w}^*) \right] \right) + \frac{\kappa_m \eta_s \rho \Gamma^2}{2} \tag{58}$$

where $\vec{w}^*$ is a solution that is as good or better than a solution at iteration $T_{\max}$ over the randomness of the algorithm. Formally it can be defined as below

**Definition D.1** (Optimal weight $\vec{w}^*$). Let $p \sim \mathbf{P}$ represent the randomness of the training algorithm

$$p \longmapsto \vec{w}_{T_{\max}}(p)$$

denote the mapping that, returns the weight vector $\vec{w}_{T_{\max}}(p)$ produced by a randomized learning algorithm at iteration $T_{\max}$.

Define the optimal seed

$$p^* \in \arg \min_p \ell\big(\vec{w}_{T_{\max}}(p)\big),$$

where $\ell(\vec{w})$ is the loss function under consideration. Finally, let

$$\vec{w}^* := \vec{w}_{T_{\max}}\big(p^*\big).$$

We call $\mathbf{w}^*$ the *optimal weight at time* $T_{\max}$: it is the weight vector (among all realizations of the randomness) that achieves the minimal loss $\ell(\cdot)$ at iteration $T_{\max}$.

We can set $\tau$ to this worst case upper bound, thus we get

$$\tau = \frac{\kappa_m}{\eta T_{z_i}} \mathbb{E}_t \left[ \ell(\vec{w}_0) - \ell(\vec{w}^*) \right] + \frac{\kappa_m \eta_s \rho \Gamma_v^2}{2} \tag{59}$$

Rearranging:

$$T_{z_i} \left[ \frac{\tau \eta}{\kappa_m} - \frac{\eta \, \eta_s \rho \Gamma_v^2}{2} \right] = \mathbb{E}_t \left[ \ell(\vec{w}_0) - \ell(\vec{w}^*) \right] \tag{60}$$

$$= \mathbb{E}_{t,z_i} \left[ \ell(\vec{w}_0, \vec{z}_i) - \ell(\vec{w}^*, \vec{z}_i) \right] \tag{61}$$

$$= \mathbb{E}_{t,z_i} \left[ \ell(\vec{w}_0, \vec{z}_i) - \ell(\vec{w}^*, \vec{z}_i) \right] \pm \mathbb{E}_{t,z_i} \left[ \ell^{\backslash z_i}(\vec{w}^*, \vec{z}_i) \right] \tag{62}$$

$$= \mathbb{E}_{t,z_i} \left[ \ell^{\backslash z_i}(\vec{w}^*, \vec{z}_i) - \ell(\vec{w}^*, \vec{z}_i) \right] + \mathbb{E}_{t,z_i} \left[ \ell(\vec{w}_0, \vec{z}_i) - \ell^{\backslash z_i}(\vec{w}^*, \vec{z}_i) \right] \tag{63}$$

Taking expectation over the randomness of the algorithm $p$ we get

$$\mathbb{E}_p[T_{z_i}] \left[ \frac{\tau \eta}{\kappa_m} - \frac{\eta \, \eta_s \rho \Gamma_v^2}{2} \right] = \mathbb{E}_{t,p,z_i} \left[ \ell^{\backslash z_i}(\vec{w}^*, \vec{z}_i) - \ell(\vec{w}^*, \vec{z}_i) \right] + \mathbb{E}_{t,z_i,p} \left[ \ell(\vec{w}_0, \vec{z}_i) - \ell^{\backslash z_i}(\vec{w}^*, \vec{z}_i) \right] \tag{64}$$

$$= \mathbb{E}_{t,z_i} \left[ L \operatorname{mem}(\vec{z}_i) \right] + \mathbb{E}_{t,z_i,p} \left[ \ell(\vec{w}_0, \vec{z}_i) - \ell^{\backslash z_i}(\vec{w}^*, \vec{z}_i) \right] \tag{65}$$

Taking expectation over $z_i$:

$$\mathbb{E}_{p,z_i}[T_{z_i}] \left[ \frac{\tau \eta}{\kappa_m} - \frac{\eta \, \eta_s \rho \Gamma_v^2}{2} \right] = \mathbb{E}_{t,z_i} \left[ L \operatorname{mem}(\vec{z}_i) \right] + \mathbb{E}_{t,z_i,p} \left[ \ell(\vec{w}_0, \vec{z}_i) - \ell^{\backslash z_i}(\vec{w}^*, \vec{z}_i) \right]$$

Using Stability (Eq. 4) we have

$$\mathbb{E}_{p,z_i}[T_{z_i}] \left[ \frac{\tau \eta}{\kappa_m} - \frac{\eta \, \eta_s \rho \Gamma_v^2}{2} \right] \geq \mathbb{E}_{z_i} \left[ L \operatorname{mem}(\vec{z}_i) \right] + \mathbb{E}_{z_i,p} \left[ \ell(\vec{w}_0, \vec{z}_i) - \ell(\vec{w}^*, \vec{z}_i) \right] - \beta$$

Since $\mathbb{E}_{p,z_i}\left[\ell(\vec{w}_0, \vec{z}_i) - \ell(\vec{w}^*, \vec{z}_i)\right] \geq 0$:

$$\mathbb{E}_{p,z_i}[T_{z_i}]\left[\frac{\tau\eta}{\kappa_m} - \frac{\eta\,\eta_s\rho\Gamma_v^2}{2}\right] \geq \mathbb{E}_{z_i}\left[L\,\mathrm{mem}(\vec{z}_i)\right] - \beta$$

Let:

$$\kappa_T = \frac{\tau\eta}{\kappa_m L} - \frac{\eta\,\eta_s\rho\Gamma_v^2}{2L}$$

Since

$$T_{z_i}\left[\frac{\tau\eta}{\kappa_m L} - \frac{\eta\,\eta_s\rho\Gamma_v^2}{2L}\right] = \frac{1}{L}\mathbb{E}_t\left[\ell(\vec{w}_0) - \ell(\vec{w}^*)\right] \geq 0$$

$$\left[\frac{\tau\eta}{\kappa_m L} - \frac{\eta\,\eta_s\rho\Gamma_v^2}{2L}\right] \geq 0, \text{ Since, } T_{z_i} \geq 1$$

Finally we have:

$$\mathbb{E}_{z_i}\left[\mathrm{mem}(\vec{z}_i)\right] \leq \kappa_T\,\mathbb{E}_{p,z_i}[T_{z_i}] + \frac{\beta}{L} \quad\blacksquare$$

### D.2.1. PROOF OF THEOREM 5.3 FOR CROSS ENTROPY LOSS

Consider Eq. 64, we have

$$\mathbb{E}_p[T_{z_i}]\left[\frac{\tau\eta}{\kappa_m} - \frac{\eta\,\eta_s\rho\Gamma_v^2}{2}\right] = \mathbb{E}_{t,p,z_i}\left[\ell^{\backslash z_i}(\vec{w}^*, \vec{z}_i) - \ell(\vec{w}^*, \vec{z}_i)\right] + \mathbb{E}_{t,z_i,p}\left[\ell(\vec{w}_0, \vec{z}_i) - \ell^{\backslash z_i}(\vec{w}^*, \vec{z}_i)\right]$$

Using the result A.4 from Ravikumar et al. (2024a) which states for cross-entropy loss we have the following result:

$$\mathbb{E}_p\left[\ell^{\backslash z_i}(\vec{w}^*, \vec{z}_i) - \ell(\vec{w}^*, \vec{z}_i)\right] \geq \mathrm{mem}(\vec{z}_i) \tag{66}$$

Taking expectation w.r.t, $\vec{z}_i$ and following the same procedure as the proof above we get: Taking expectation over $z_i$:

$$\mathbb{E}_{p,z_i}[T_{z_i}]\left[\frac{\tau\eta}{\kappa_m} - \frac{\eta\,\eta_s\rho\Gamma_v^2}{2}\right] \geq \mathbb{E}_{t,z_i}\left[\mathrm{mem}(\vec{z}_i)\right] + \mathbb{E}_{t,z_i,p}\left[\ell(\vec{w}_0, \vec{z}_i) - \ell^{\backslash z_i}(\vec{w}^*, \vec{z}_i)\right]$$

Using Stability (Eq. 4) we have

$$\mathbb{E}_{p,z_i}[T_{z_i}]\left[\frac{\tau\eta}{\kappa_m} - \frac{\eta\,\eta_s\rho\Gamma_v^2}{2}\right] \geq \mathbb{E}_{z_i}\left[\mathrm{mem}(\vec{z}_i)\right] + \mathbb{E}_{z_i,p}\left[\ell(\vec{w}_0, \vec{z}_i) - \ell(\vec{w}^*, \vec{z}_i)\right] - \beta$$

Since $\mathbb{E}_{p,z_i}\left[\ell(\vec{w}_0, \vec{z}_i) - \ell(\vec{w}^*, \vec{z}_i)\right] \geq 0$:

$$\mathbb{E}_{p,z_i}[T_{z_i}]\left[\frac{\tau\eta}{\kappa_m} - \frac{\eta\,\eta_s\rho\Gamma_v^2}{2}\right] \geq \mathbb{E}_{z_i}\left[\mathrm{mem}(\vec{z}_i)\right] - \beta$$

Let:

$$\kappa_{TX} = \frac{\tau\eta}{\kappa_m} - \frac{\eta\,\eta_s\rho\Gamma_v^2}{2}$$

Since

$$T_{z_i}\left[\frac{\tau\eta}{\kappa_m} - \frac{\eta\,\eta_s\rho\Gamma_v^2}{2}\right] = \mathbb{E}_t\left[\ell(\vec{w}_0) - \ell(\vec{w}^*)\right] \geq 0$$

$$\left[\frac{\tau\eta}{\kappa_m} - \frac{\eta\,\eta_s\rho\Gamma_v^2}{2}\right] \geq 0, \text{ Since, } T_{z_i} \geq 1$$

Finally we have:

$$\mathbb{E}_{z_i}\left[\mathrm{mem}(\vec{z}_i)\right] \leq \kappa_{TX}\,\mathbb{E}_{p,z_i}[T_{z_i}] + \beta \quad\blacksquare$$

This is the same result as Theorem 5.3 but with $L = 1$.

### D.2.2. REMARK ON THE REQUIREMENT FOR $\beta$-STABILITY

Consider Eq. 64. We have:

$$\mathbb{E}_p[T_{z_i}] \left[ \frac{\tau\eta}{\kappa_m} - \frac{\eta\,\eta_s\rho\Gamma_v^2}{2} \right] = \mathbb{E}_{t,p,z_i} \left[ \ell^{\backslash z_i}(\vec{w}^*, \vec{z}_i) - \ell(\vec{w}^*, \vec{z}_i) \right]$$
$$+ \mathbb{E}_{t,z_i,p} \left[ \ell(\vec{w}_0, \vec{z}_i) - \ell^{\backslash z_i}(\vec{w}^*, \vec{z}_i) \right].$$

Assuming sufficient generalization, we can reasonably state that:

$$\mathbb{E}_{t,z_i,p} \left[ \ell(\vec{w}_0, \vec{z}_i) - \ell^{\backslash z_i}(\vec{w}^*, \vec{z}_i) \right] \geq 0.$$

Thus, we can write:

$$\mathbb{E}_p[T_{z_i}] \left[ \frac{\tau\eta}{\kappa_m} - \frac{\eta\,\eta_s\rho\Gamma_v^2}{2} \right] \geq \mathbb{E}_{t,p,z_i} \left[ \ell^{\backslash z_i}(\vec{w}^*, \vec{z}_i) - \ell(\vec{w}^*, \vec{z}_i) \right],$$
$$\mathbb{E}_p[T_{z_i}] \left[ \frac{\tau\eta}{\kappa_m} - \frac{\eta\,\eta_s\rho\Gamma_v^2}{2} \right] \geq \mathbb{E}_{t,z_i} \left[ \mathrm{mem}(\vec{z}_i) \right]$$

Finally, we conclude:

$$\mathbb{E}_{t,z_i} \left[ \mathrm{mem}(\vec{z}_i) \right] \leq \kappa_T \, \mathbb{E}_p[T_{z_i}]$$

Thus we can drop the requirement for stability.

### D.3. CSL and Learning Time

**Proof of Theorem 5.4**

**Assumptions**:

- Assume that the loss function $\ell$ is $\rho$-Lipschitz.

- Assume that the gradient estimator $\widetilde{\nabla}_w\ell$ is unbiased i.e., $\mathbb{E}_t[\widetilde{\nabla}_w\ell(\vec{w}_t)] = \nabla_w\ell(\vec{w}_t)$

- Assume bounded gradient as stated in Assumption 3.

**Proof**

We know from Equation 59

$$\tau = \frac{\kappa_m}{\eta T_{z_i}} \mathbb{E}_t \left[ \ell(\vec{w}_0) - \ell(\vec{w}_{T_{z_i}}) \right] + \frac{\kappa_m\eta_s\rho\Gamma_v^2}{2}$$

Rearranging:

$$T_{z_i} \left[ \frac{\tau\eta}{\kappa_m} - \frac{\eta\,\eta_s\rho\Gamma_v^2}{2} \right] = \mathbb{E}_t \left[ \ell(\vec{w}_0) - \ell(\vec{w}_{T_{z_i}}) \right]$$

$$T_{z_i} \left[ \frac{\tau\eta}{\kappa_m} - \frac{\eta\,\eta_s\rho\Gamma_v^2}{2} \right] \leq T_{\max} \left[ \frac{\tau\eta}{\kappa_m} - \frac{\eta\,\eta_s\rho\Gamma_v^2}{2} \right] \leq \sum_{t=0}^{T_{\max}-1} \mathbb{E}_t \left[ \ell(\vec{w}_t) - \ell(\vec{w}_{t+1}) \right] \qquad \leq L$$

$$\leq \sum_{t=0}^{T_{\max}-1} \mathbb{E}_t \left[ \ell(\vec{w}_t) \right] - \sum_{t=0}^{T_{\max}-1} \mathbb{E}_t \left[ \ell(\vec{w}_{t+1}) \right] \qquad \leq L$$

$$\leq \sum_{t=0}^{T_{\max}-1} \mathbb{E}_{t,z_i} \left[ \ell(\vec{w}_t, \vec{z}_i) \right] - \sum_{t=0}^{T_{\max}-1} \mathbb{E}_{t,z_i} \left[ \ell(\vec{w}_{t+1}, \vec{z}_i) \right] \qquad \leq L$$

Taking expectation over $z_i$ we get:

$$\mathbb{E}_{z_i}\left[T_{z_i}\right]\left[\frac{\tau\eta}{\kappa_m} - \frac{\eta\,\eta_s\rho\Gamma_v^2}{2}\right] \leq \mathbb{E}_{t,z_i}\left[\sum_{t=0}^{T_{\max}-1}\ell(\vec{w}_t,\vec{z}_i)\right] - \mathbb{E}_{t,z_i}\left[\sum_{t=0}^{T_{\max}-1}\ell(\vec{w}_{t+1},\vec{z}_i)\right] \qquad \leq L$$

$$\leq \mathbb{E}_{t,z_i}\left[\text{CSL}(\vec{z}_i)\right] - \mathbb{E}_{t,z_i}\left[\sum_{t=0}^{T_{\max}-1}\ell(\vec{w}_{t+1},\vec{z}_i)\right] \qquad \leq L$$

To identify a tight bound that is also more interpretable, we identify $\xi$ as below:

$$\xi = \mathbb{E}_{t,z_i}\left[\sum_{t=0}^{T_{\max}-1}\ell(\vec{w}_{t+1},\vec{z}_i)\right]$$

Thus $\xi$ can be interpreted as a offset to scale CSL correctly. Putting this together we have

$$L\kappa_T\,\mathbb{E}_{z_i}\left[T_{z_i}\right] \leq \mathbb{E}_{t,z_i}\left[\text{CSL}(\vec{z}_i)\right] - \xi \leq L$$

$$\kappa_T\,\mathbb{E}_{z_i}\left[T_{z_i}\right] \leq \frac{\mathbb{E}_{t,z_i}\left[\text{CSL}(\vec{z}_i)\right] - \xi}{L} \leq 1 \quad \blacksquare \tag{67}$$

### D.3.1. NOTE ON UNBOUNDED LOSS

Note since the $L$-bounded loss is only used for the upper bound, this result hold true for cross entropy, but with out the upper bound, i.e.

$$\kappa_{TX}\,\mathbb{E}_{z_i}\left[T_{z_i}\right] \leq \mathbb{E}_{t,z_i}\left[\text{CSL}(\vec{z}_i)\right] - \xi \quad \blacksquare \tag{68}$$

### D.3.2. $\xi$ - A LOWER BOUND FOR CSL

In this section, we build intuition on $\xi$. We will see that $\xi$ can be interpreted as a lower bound on CSL, which scales CSL into the appropriate range to bound learning time and memorization.

Consider $\xi$ as defined by

$$\xi = \mathbb{E}_{t,z_i}\left[\sum_{t=0}^{T_{\max}-1}\ell(\vec{w}_{t+1},\vec{z}_i)\right]$$

Now assume $\mathbb{E}_{z_i}[\ell(\vec{w}_0,\vec{z}_i)] \geq \mathbb{E}_{z_i}[\ell(\vec{w}_{T_{\max}},\vec{z}_i)]$. This assumption, which is empirically justified for most deep neural networks, states that the loss at the beginning of training is higher than at the end. Under this assumption, we obtain:

$$\xi = \mathbb{E}_{t,z_i}\left[\sum_{t=0}^{T_{\max}-1}\ell(\vec{w}_{t+1},\vec{z}_i)\right]$$

$$= \mathbb{E}_{t-1,z_i}\left[\sum_{t=1}^{T_{\max}-1}\ell(\vec{w}_{t+1},\vec{z}_i) + \ell(\vec{w}_{T_{\max}},\vec{z}_i)\right]$$

$$\leq \mathbb{E}_{t-1,z_i}\left[\sum_{t=1}^{T_{\max}-1}\ell(\vec{w}_{t+1},\vec{z}_i) + \ell(\vec{w}_0,\vec{z}_i)\right]$$

$$\leq \mathbb{E}_{t-1,z_i}\left[\sum_{t=0}^{T_{\max}-1}\ell(\vec{w}_{t+1},\vec{z}_i)\right]$$

$$\xi \leq \mathbb{E}_{t-1,z_i}\left[\text{CSL}(\vec{z}_i)\right]$$

Since this holds for any $\vec{z}_i \sim \mathbf{Z}$, we conclude that $\xi$ is a lower bound on CSL. Moreover, when considering the result for memorization (see Theorem 5.5), we observe that $\xi$ scales CSL so that it lies in the range $\approx (0,1)$ when $\beta \approx 0$.

## D.4. Memorization and Loss

**Proof of Theorem 5.5**

**Assumptions:**

- Assume that the loss function $\ell$ is $\rho$-Lipschitz.

- Assume that the gradient estimator $\widetilde{\nabla}_w \ell$ is unbiased i.e., $\mathbb{E}_t[\widetilde{\nabla}_w \ell(\vec{w}_t)] = \nabla_w \ell(\vec{w}_t)$.

- Assume bounded gradient as stated in Assumption 3.

- Assume loss is bounded $0 \leq \ell \leq L$

- Assume $\beta$-stability as stated in Assumption 4.

**Proof**

This proof results from Theorem 5.3 and taking expectation over $p$ (randomness of the algorithm) on Theorem 5.4 we have:

$$\mathbb{E}_{z_i}\left[\mathrm{mem}(\vec{z}_i)\right] \leq \frac{\mathbb{E}_{z_i,p}\left[\mathrm{CSL}(\vec{z}_i)\right] - \xi + \beta}{L} \leq 1 + \frac{\beta}{L} \quad \blacksquare \tag{69}$$

### D.4.1. NOTE ON CROSS ENTROPY LOSS AND NON-TRIVAL BOUND

Here we use the cross entropy results from Theorem 5.3 and 5.4. Specifically, using Equations 67 and 68. We take expectation over the randomness of the algorithm $p$ on Equation 68 and use this in Eq. 67 to get

$$\mathbb{E}_{z_i}\left[\mathrm{mem}(\vec{z}_i)\right] \leq \mathbb{E}_{z_i,p}\left[\mathrm{CSL}(\vec{z}_i)\right] + \beta - \xi \quad \blacksquare \tag{70}$$

**Non-triviality.** Here we would like to emphasize that the CSL bound on memorization is non-trivial. Specifically consider the result from Appendix D.2.2. Re-stating the result for convenience we get:

$$\mathbb{E}_{t,z_i}\left[\mathrm{mem}(\vec{z}_i)\right] \leq \kappa_T \, \mathbb{E}_p[T_{z_i}]$$

This is under the reasonable generalization assumption:

$$\mathbb{E}_{t,z_i,p}\left[\ell(\vec{w}_0, \vec{z}_i) - \ell^{\backslash z_i}(\vec{w}^*, \vec{z}_i)\right] \geq 0.$$

Using this result in the proof of Theorem 5.5 we get:

$$\mathbb{E}_{z_i}\left[\mathrm{mem}(\vec{z}_i)\right] \leq \frac{\mathbb{E}_{z_i,p}\left[\mathrm{CSL}(\vec{z}_i)\right] - \xi}{L} \leq 1 \quad \blacksquare \tag{71}$$

Thus CSL is effectively bounded between the memorization score and the trivial upper bound (i.e. 1), showing the CSL bound is non-trivial.

