# OpenReview forum: "Towards Memorization Estimation: Fast, Formal and Free"
_ICML.cc/2025/Conference — ICML 2025 poster_

### Official Review · Reviewer_fZEA · 2025-03-05

**Overall Recommendation:** 4

**Summary:**

This paper introduces cumulative sample loss, CSL as a way to measure memorization in neural networks. The key idea is that by tracking the cumulative loss over training, CSL can identify mislabeled and duplicate examples efficiently. The authors argue that CSL is both cheaper than stability-based methods and more effective at finding problematic examples. They evaluate their method on standard benchmark datasets and show that it outperforms existing heuristics for detecting mislabeled data.

**Claims And Evidence:**

The authors claim that CSL is a better measure of memorization than previous methods, particularly stability-based approaches. The experimental results generally support this, showing that CSL can effectively finds mislabeled and duplicate samples. But, the claims about general memorization seem a bit narrow becuase the paper mainly focuses on mislabeled examples and does not explore broader notions of memorization, such as how it relates to generalization or robustness. Some comparisons to existing memorization measures could be stronger, especially with alternative ways of grouping examples based on learning dynamics.

**Essential References Not Discussed:**

There are a few missing references that seem very relevant to this work. For example:

https://arxiv.org/abs/2202.09931 studies learning dynamics as a way to classify examples into different categories. At the core this is very similar to CSL.
https://arxiv.org/abs/2412.07684 studies the connection between memorization and generalization.
https://arxiv.org/pdf/2207.00099 and https://arxiv.org/pdf/2309.16748 also look at memorization from a learning dynamics perspective.

**Experimental Designs Or Analyses:**

The experiments are well-structured and support the main claims of the paper. The ablation studies show that CSL is effective at detecting mislabeled data, and the comparison with stability-based methods highlights its efficiency. However, the evaluation could be more comprehensive. For example, it would be useful to test CSL across different architectures or dataset sizes to see if the trends hold more broadly. Additionally, it would be good to analyze whether CSL correlates with other memorization-related signals, such as those used in curriculum learning or adversarial robustness.

**Methods And Evaluation Criteria:**

The method itself is quite simple: the authors track the cumulative loss over training and use this as a signal for memorization. The evaluation criteria, identifying mislabeled and duplicate samples make sense for the problem, but they don’t fully capture memorization in a more general sense. It would be good to see how CSL performs on other memorization-related tasks, such as distinguishing between easy-to-learn and hard-to-learn examples or detecting spurious correlations. The benchmarks used are reasonable, but additional datasets or different noise levels could provide a more complete picture.

**Other Comments Or Suggestions:**

- It would be helpful to clarify whether CSL is robust across different architectures or if it mainly works well for the specific models tested.
- Some additional theoretical grounding for why CSL is expected to work better than other methods would strengthen the argument.

**Other Strengths And Weaknesses:**

Strengths:
- CSL is a very simple and computationally cheap approach, making it practical for real-world use.
- The method effectively detects mislabeled and duplicate examples, which is useful for dataset curation, specially these days that curating datasets seems more important than ever.
- The paper is well-written and easy to follow.

Weaknesses:
- The focus on mislabeled examples is somewhat narrow and does not fully capture the complexity of memorization.
- The comparisons to existing memorization measures could be stronger, especially with alternative approaches that the effect of memorization on generalization.

**Questions For Authors:**

1. How does CSL compare to other learning-dynamics-based methods beyond stability-based memorization?
2. Have you tried applying CSL to a setting where memorization is not just about mislabeled examples but also about spurious correlations? For example memorization of examples that all benefit from a spurious correlation.

---
# Update on Mar 31st:

Dear authors,

At AC's request, here I provide more details on my initial review. Apologies for any missing details. I would be happy to update my score once I receive your reply.

## Which other architectures and dataset sizes?
This paper reports results using a ResNet-18. But they reports the results against the memorization scores precomputed in [1] which was using a ResNet-50. So it makes sense to report on a ResNet-50 with the exact setup described in Appendix B of [1].

Running experiments on modern architectures like ViTs, specially since they have different inductive biases compared to ResNets, would also strengthen the paper. But I don’t expect the authors to rerun all their experiments with ViTs.

Let me elaborate why I wished to see further experiment: Another aspect of memorization, beyond mislabeled examples, is the memorization of underrepresented examples. For example, if you train a large model on just a handful of examples, the model gets zero training error but won’t generalize. According to the definition of the memorization score in [1] (the difference between held-in and held-out performance), this is memorization. It would be interesting to test whether CSL can also identify memorized (but not mislabeled) examples in such cases. A natural testbed would be long-tailed datasets like ImageNet-LT or CIFAR-LT, where certain classes are underrepresented.

## Other memorization-related signals used in curriculum learning or adversarial robustness:
In curriculum learning, [2] uses forgetting frequency as a signal: how often a sample flips from being correctly to incorrectly classified. [3] uses loss trajectory clustering to group examples similar to CSL. [4] uses model confidence on the true class and the variability of that confidence across epochs.

In adversarial robustness, [5] proposes loss-sensitivity, suggesting that memorized examples are more easily perturbed. [6] uses sharpness of the loss function at convergence as another signal.

## Other learning-dynamics-based methods:
Each of the papers above suggests a different signal: loss sensitivity, forgetting frequency, confidence, or confidence variability that may be compared with CSL. A comparison to at least one of these could strengthen the paper.

## The theory and the supplementary material:
I checked the supplementary material. I had initially missed that the authors also conducted experiments with VGG, MobileNetV2, and Inception. However, since the ground-truth memorization scores are precomputed using a ResNet-50, it would make a lot of sense to include that for a more direct comparison.

I also revisited the theoretical section in the appendix and realized I had initially missed the link the authors make between two parts: one showing that CSL bounds learning time, and the other that learning time bounds memorization. I now value the theoretical contribution more.


Thank you!

[1] "What neural networks memorize and why: Discovering the long tail via influence estimation" Vitaly Feldman and Chiyuan Zhang. NeurIPS 2020.

[2] "An empirical study of example forgetting during deep neural network learning" Toneva et al. ICLR 2019.

[3] "Deconstructing Distributions: A Pointwise Framework of Learning" Kaplun et al. ICLR 2023.

[4] "Dataset Cartography: Mapping and Diagnosing Datasets with Training Dynamics. Swayamdipta et al. EMNLP 2020.

[5] "A Closer Look at Memorization in Deep Networks" Arpit et al. ICML 2017.

[6] "Deep Nets Don't Learn via Memorization" Krueger et al. ICLR 2017.

**Relation To Broader Scientific Literature:**

The paper is connected to prior work on memorization in deep learning, particularly methods that use learning dynamics to classify examples as memorized or generalized. However, it focuses mainly on mislabeled and duplicate examples, which is only one aspect of memorization. There are several recent works that explore similar ideas but in a broader context, including those that analyze memorization in terms of generalization or in terms of progressive learning dynamics. A more thorough discussion of how CSL fits within these broader frameworks would improve the positioning of the paper.

**Theoretical Claims:**

There aren’t many deep theoretical results in the paper, but the reasoning behind CSL is intuitive and aligns with prior work on learning dynamics. The authors suggest that CSL correlates with memorization, but this is mostly supported empirically rather than through a formal analysis.

---

> ### Author Rebuttal · Authors · 2025-03-31
>
> We thank the reviewer for their valuable feedback, we address their questions below. Tables and Figures are provided at ****[Rebuttal Page clickable link](https://lively-dune-08c51d610.6.azurestaticapps.net/).****
>
> 1. Which other architectures and dataset sizes?
>
> **A**: Please see our response to reviewer ZrxQ Q1. Additionally we also ran the the long tail experiments, visualized in Figure 3,4,5,6 in the rebuttal page. See the Q2 for a more in-depth explanation.
>
> 2. Other memorization-related signals used in curriculum learning or adversarial robustness:
>
> **A**: We would like to clarify that we provide two experiments, first *mislabeled detection*, second *duplicate detection* to capture mem. properties. Additionally, we ran the exps below
>
> - Adversarial Robustness (Fig 7 rebuttal page):
>     - We provide a visualization of  adversarial distance versus memorization  and CSL scores. Adversarial distance refers to how easily a sample can be perturbed to change its classification. The results show that samples with low adversarial distance (more vulnerable) tend to have higher memorization and CSL scores. This suggests that CSL and memorization capture similar properties and are strongly related to model robustness.
> - Easy vs. Hard-to-Learn Samples:
>     - Fig 1 (on the Rebuttal Page (link above)) shows images with low CSL scores from CIFAR100, interpreted as easy or typical examples (class prototypes). Fig. 2 shows high CSL images from CIFAR100 capturing atypical and likely memorized examples.  Fig 3,4 presents the same for ImageNet. The high CSL visualizations clearly show we identify memorized (but not mislabeled) examples in such cases. This visualization is similar to [3] who show similar figures with their proposed proxy of curvature.
> - Long Tail Behavior
> 	-  Clean CIFAR100 and ImageNet datasets are also long tailed [11]. Fig 5 and Fig 6 plots the histogram of CSL on CIFAR100 and ImageNet showing that CSL captures long tail behavior
> - Additional Results Under Higher Noise (Table 2 rebuttal page)
>     - We evaluate CSL under higher label noise settings.
>     - CSL maintains its ability to distinguish clean from mislabeled examples, indicating its robustness across different noise levels.
>
> 3. On Other learning dynamics method?
>
> **A**: Please note, Second Split Forgetting time (SSFT), Forgetting time, learning time, in-confidence (1 - confidence) and Curvature are learning dynamics based methods and clearly we see that CSL out performs them in both mislabeled detection and duplicate detection (Tables 2 and 3 in the main paper).  Regarding similarity with memorization scores we have added forgetting frequency, loss sensitivity, final epoch loss to the (Table 1 rebuttal page) as additional baselines.
>
> 4. Supplementary Material
>
> **A**: Feldman and Zhang [11] ran Inception on CIFAR10/100 and ResNet50 on ImageNet. Additionally, we have added ResNet50 to ImageNet experiments (Table 1 rebuttal page) and Inception on CIFAR100. To save space we kindly ask the reviewer to see our response to reviewer ZrxQ Q1.
>
> 5. There are several recent works that explore similar ideas but in a broader context,.. would improve the positioning of the paper.
>
> **A**: Thank you for bringing this to our attention. While these works share similar goals, they differ from ours. We will include them in the related work section. [7] measures performance across multiple models on a single input, while CSL analyzes the distribution of memorization scores within a single training run, with no added overhead. In contrast, [7] is significantly more computationally expensive. [8] addresses how memorization and spurious correlations hinder generalization and proposes MAT to mitigate them. Our work introduces CSL, a fast, theoretical metric for estimating memorization, which we apply to detect mislabeled and duplicate data. [9] explores how memorization fades when examples are removed. CSL, on the other hand, focuses on efficiently estimating how memorized each sample is, offering both theoretical grounding and empirical utility. [10] proposes data splitting for robust OOD training. CSL instead offers a lightweight and accurate method for measuring memorization. Additionally please see our response to reviewer WoPj Q5.
>
> [3] Garg et al. "Memorization Through the Lens of Curvature of Loss Function Around Samples." ICML 24.\
> [7] Kaplun et al. "Deconstructing Distributions: A Pointwise Framework of Learning." ICLR 23\
> [8] Bayat et al.  "The Pitfalls of Memorization: When Memorization Hurts Generalization." ICLR 25\
> [9] Jagielski et al. "Measuring Forgetting of Memorized Training Examples." ICLR 23\
> [10] Pezeshki, et al. "Discovering Environments with XRM." ICML 24\
> [11] Feldman & Zhang  "What neural networks memorize and why: Discovering the long tail via influence estimation" NeurIPS 20.

---

> > ### Comment · Reviewer_fZEA · 2025-04-03
> >
> > I read your rebuttal and the additional experiments and analysis that you have provided. I must say that I am impressed with the effort you have put into addressing my previous concerns.
> > The new experiments and analysis have strengthened the paper and I would increase my score.

---

### Official Review · Reviewer_woPj · 2025-03-11

**Overall Recommendation:** 4

**Summary:**

The paper proposes a computational efficient proxy metric (CSL) to the popular notion of memorization proposed by Zhang and Feldman (2020). The authors support this metric with theoretical analyses and empirical results on standard image classification benchmarks.

**Claims And Evidence:**

The empirical claims that CSL are a better proxy metric to the baselines are convincing. There are 3 main portions of the empirical results:

1. CSL is better correlated with memorization than the curvature-based method and is also more computationally efficient.
2. CSL also performs better at detecting mislabeled samples due to symmetric label noise than the baselines.
3. CSL detects duplicate samples with higher accuracy than the baselines.

**Essential References Not Discussed:**

Not that I know of.

**Experimental Designs Or Analyses:**

Question: Is there a particular reason for choosing cosine similarity as a metric instead of Pearson's correlation coefficient or mean square error? Correlation coefficient seems the most natural to me.

**Methods And Evaluation Criteria:**

Experiment methods and evaluation criteria are reasonable to me.

**Question**: There is no description of how CSL is used to detect mislabeled samples (Section 7.1) and duplicate samples (Section 7.2). I also cannot find it in Appendix B.1 or B.5. Is this simply thresholding the CSL for each sample?

**Other Comments Or Suggestions:**

The paper is very well-written in my opinion. Even as a non-theorist, I feel like I understand the theorems and learn some useful tricks along the way. I feel like Lemma 5.1 can be generally useful, and it’d be a nice contribution too (if it’s not been shown before). The experiments are also convincing and a nice addition to the theoretical results.

**Other Strengths And Weaknesses:**

N/A

**Questions For Authors:**

Question: I could not find how $\ell^{\backslash z_i}$ is defined. Is it loss when the sample $z_i$ is removed from the training set?

**Relation To Broader Scientific Literature:**

I believe that related works have been mentioned throughout the paper. I personally would like to see more discussion of the prior literature in more details.

**Theoretical Claims:**

I skimmed over the proofs but did not check them carefully. All the claims and theorems seem to be reasonable, but I do have some questions below.

**The expectation over samples.** Starting from Theorem 5.3, there is an introduction to an expectation over training samples ($z_i$). Looking at the proof quickly I believe that it is an expectation over the entire training data distribution. This already makes me curious about how this theorem would be used because we often care about memorization of a particular sample, not an expectation over a distribution.

However, on L246, Theorem 5.3 is being interpreted as applying to “a group of samples” like $U(T)$ which is a subset of all training data $S$. It is a bit unclear to me why this interpretation is valid, given the theorem holds in expectation for the entire training distribution. This is fairly important as it is used to motivate the experiments showing relationship between CSL and “Mem Score” (Figure 6 and 8). I might be missing something here.

**Assumption on L257**: “it can be assumed that $\kappa_T$ is constant across different subsets $U(T)$.” This assumption may need more explanation for why it is reasonable. My guess is that because $k_g$ depends on Frobenius norm and the singular values of the samples batch, which are arguably irrelevant to the difficulty to learn by neural network.This assumption may need more explanation for why it is reasonable. My guess is that because $k_g$ depends on Frobenius norm and the singular values of the samples batch, which are arguably irrelevant to the difficulty to learn by neural networks.

---

> ### Author Rebuttal · Authors · 2025-03-31
>
> We thank the reviewer for their valuable feedback, we address their questions below. Tables and Figures are provided at ****[Rebuttal Page clickable link](https://lively-dune-08c51d610.6.azurestaticapps.net/).****
>
> 1. There is no description of how CSL is used to detect mislabeled samples (Section 7.1) and duplicate samples (Section 7.2). I also cannot find it in Appendix B.1 or B.5. Is this simply thresholding the CSL for each sample?
>
> **A**: Thank you for bringing this to our attention, we will clarify this in the revision. We do indeed use thresholding on each sample’s CSL, giving us a binary detector whose performance is measured and reported using AUROC.
>
> 2. The expectation over samples. Starting from Theorem 5.3, there is an introduction to an expectation over training samples (). Looking at the proof quickly I believe that it is an expectation over the entire training data distribution. This already makes me curious about how this theorem would be used because we often care about memorization of a particular sample, not an expectation over a distribution. However, on L246, Theorem 5.3 is being interpreted as applying to “a group of samples” like  which is a subset of all training data . It is a bit unclear to me why this interpretation is valid, given the theorem holds in expectation for the entire training distribution. This is fairly important as it is used to motivate the experiments showing relationship between CSL and “Mem Score” (Figure 6 and 8). I might be missing something here.
>
> **A**: We would like to clarify that we meant arbitrary **random** subsets. Consider selecting a subset U at random; then T can be set as $\max T_{z_i}$ for $z_i \in U$. Since Equation (10) holds for such a distribution, we obtain the stated result. We will revise the text to clarify this point and explicitly indicate that the subsets under consideration are random.
>
> 3. Assumption on L257: “it can be assumed that is constant across different subsets .” This assumption may need more explanation for why it is reasonable. My guess is that because depends on Frobenius norm and the singular values of the samples batch, which are arguably irrelevant to the difficulty to learn by neural network. This assumption may need more explanation for why it is reasonable. My guess is that because depends on Frobenius norm and the singular values of the samples batch, which are arguably irrelevant to the difficulty to learn by neural networks.
>
> **A**: The reviewers explanation is intuitive and correct. More formally, note that $\kappa_g$  is not dependent on the training mini batch but is dependent on the choice of input made for computing the gradient norm w.r.t to the input, which is user controlled. The the input gradient bound for a given input is independent of the choice of the training mini batch. Now choosing $\max_u \kappa_{g_u},~u \subset S$, i.e. max value over the subsets, we can get a worst case upper bound constant.
>
> 4. Is there a particular reason for choosing cosine similarity as a metric instead of Pearson's correlation coefficient or mean square error? Correlation coefficient seems the most natural to me.
>
> **A**: We used cosine similarity because it was employed in prior methods and setups [3]. We have also added Pearson correlation in Table 1 (see the rebuttal page linked above). Any magnitude greater than 0.25 is considered a statistically significant correlation. Thus, the takeaways are similar between cosine similarity and Pearson correlation. Please also see our response to Q1 for reviewer ZrxQ.
>
> 5. I believe that related works have been mentioned throughout the paper. I personally would like to see more discussion of the prior literature in more details.
>
> **A**: Thank you for the suggestion, other reviewers have also expressed similar concerns. We will expand our Related Works section to discuss more thoroughly (1) stability-based memorization and its computational challenges; (2) learning dynamics; and (3) broader loss trajectory works for membership inference attacks. This should clarify further how our approach (CSL) integrates with and advances prior research.
>
> 6. I feel like Lemma 5.1 can be generally useful, and it’d be a nice contribution too (if it’s not been shown before)
>
> **A**: Thank you for recognizing the broader utility of Lemma 5.1. To our knowledge, it has not been explicitly presented before. We will emphasize this contribution and place it more prominently in the revised manuscript.
>
> 7. I could not find how  l^z is defined. Is it loss when the sample z is removed from the  training set?
>
> **A**:  Yes, $l^{\setminus {z_i}}(z_i)$ is the per-sample loss for ($z_i$) under the model trained on the leave-one-out dataset ($S^{\setminus {z_i}}$). We will clarify that notation explicitly in the revised manuscript.
>
> [3] Garg et al. "Memorization Through the Lens of Curvature of Loss Function Around Samples." ICML 24.

---

> > ### Comment · Reviewer_woPj · 2025-04-01
> >
> > I thank the authors for their clarifications and extra experiments. I did not notice that the authors originally used ResNet-18 which does not match the pre-computed results in Feldman & Zhang, but that seems to be fixed now thanks to the other reviewers. As a result, I maintain my original rating.

---

### Official Review · Reviewer_ZrxQ · 2025-03-14

**Overall Recommendation:** 3

**Summary:**

This paper introduces Cumulative Sample Loss (CSL) as a novel proxy for measuring memorization in deep learning models. The authors formally adopt the memorization definition established by Feldman(2020); Feldman & Zhang (2020) and develop a theoretical framework connecting CSL to both training time and memorization. Through the theoretical analysis, they prove these connections under specific assumptions about learning dynamics.

The authors empirically validate their theoretical findings by demonstrating high correlation between CSL and established memorization scores across multiple datasets and model architectures. They further showcase the practical utility of CSL by applying it to detect mislabeled examples and duplicate samples in datasets, where it achieves state-of-the-art performance compared to existing methods while being computationally more efficient.

## Update after rebuttal

Most of my concerns have been adequately addressed by the authors.
1. Authors provided the calculations matching the setup from Feldman & Zhang
2. Authors clarifies the generality of the proofs
3. Authors included missing citations and clarified the relation to the prior literature

I have therefore increased my score 2->3. Some remaining concerns I have:
1. Given the training setup mismatch between some early experiments and Feldman & Zhang, I don't think it's entirely valid to use precomputed scores from the prior work - those need to be re-caclucated for each training setup
2. Given existing literature on using loss trajectories to assess vulnerability, I'd expect more extensive comparison to baselines

**Claims And Evidence:**

Discussed below

**Essential References Not Discussed:**

The paper overlooks two works with a very similar idea of using loss trajectory to assess memorization/vulnerability of the target point. The papers in question, however, frame it in the context of a Membership Inference Attack, but it's still highly relevant to the proposed research.

[1] Liu, Yiyong, et al. "Membership inference attacks by exploiting loss trajectory." _Proceedings of the 2022 ACM SIGSAC Conference on Computer and Communications Security_. 2022.

[2] Li, Hao, et al. "Seqmia: Sequential-metric based membership inference attack." _Proceedings of the 2024 on ACM SIGSAC Conference on Computer and Communications Security_. 2024.

Loss trajectory (specifically mean loss) has also been previously used as memorization proxy in privacy auditing literature, e.g.

[3] Nasr, Milad, et al. "Adversary instantiation: Lower bounds for differentially private machine learning." 2021 IEEE Symposium on security and privacy (SP). IEEE, 2021.

**Experimental Designs Or Analyses:**

The paper's experimental methodology exhibits some significant limitations that undermine the strength of its empirical validation. A primary concern is the authors' use of memorization scores precomputed by Feldman & Zhang (2020) while employing different model architectures in their own experiments (ResNet50 vs ResNet18). Since memorization scores are highly specific to particular training configurations (including architecture, optimization settings, and data processing), this mismatch raises questions about the validity of the correlation analysis between CSL and the referenced memorization scores.

Furthermore, the paper lacks comprehensive comparison against simpler baseline metrics. Examining Figure 2, it appears that final sample loss might achieve similar correlation with memorization as the proposed CSL metric. This observation suggests that accumulating loss throughout training may not provide substantial additional benefit. Without explicitly comparing against such straightforward alternatives, the paper fails to convincingly demonstrate the unique value of CSL over simpler approaches.

**Methods And Evaluation Criteria:**

The paper utilizes standard benchmarks (CIFAR-10/100, ImageNet) and common architectures following established protocols from prior work. Their evaluation employs appropriate metrics: cosine similarity for correlating CSL with memorization scores and AUROC for mislabeled/duplicate detection tasks.

A notable limitation is limited baselines for correlation with the memorization scores. While CSL outperforms input loss curvature, more extensive set of baselines is necessary - covering at least simple metrics like final loss or model confidence.

**Other Comments Or Suggestions:**

N/A

**Other Strengths And Weaknesses:**

N/A

**Questions For Authors:**

N/A

**Relation To Broader Scientific Literature:**

This paper makes a valuable contribution to the theoretical understanding of memorization in deep learning by establishing connections between CSL, learning time, and memorization. Its most significant practical contribution is providing a computationally efficient proxy for memorization, which traditionally requires expensive leave-one-out training procedures.

By demonstrating that CSL can be obtained with zero additional computational overhead during training, the authors offer a practical tool for analyzing memorization at scale. The work effectively builds upon prior research on input loss curvature while offering substantial computational advantages.

**Theoretical Claims:**

The core contribution of this paper lies in its theoretical framework connecting Cumulative Sample Loss (CSL) to memorization and learning time. However, several aspects of these theoretical claims are not entirely clear to me.

Lemma 5.1, which states that input gradient norm is bounded by weight gradient norm, appears to claim applicability to "any neural network." However, its proof in C1 only covers a linear layer with particular weight matrix configurations and with no bias.

The theoretical framework relies on several strong assumptions that may not hold in practical deep learning settings. These include bounded loss functions, smooth loss landscapes, and uniform stability guarantees (which essentially implies bounded memorization by definition). While these assumptions facilitate mathematical analysis, they potentially limit the applicability of the results to real-world deep learning scenarios where loss functions may be unbounded (e.g., cross-entropy without clipping) and loss landscapes are known to be highly non-convex and irregular.

Additionally, the paper is unclear about how the theoretical results extend to minibatch training, which is standard practice. The proofs appear to consider individual sample updates, leaving questions about how gradient interactions within minibatches might affect the derived bounds.

I also don't understand the authors' claim that certain theorems hold for any arbitrary subset of training data. This interpretation is not clearly justified, as the optimization steps in the proofs seem to rely on specific data distributions. The paper would benefit from clarifying how the expectation-based results generalize to arbitrary data subsets.

---

> ### Author Rebuttal · Authors · 2025-03-31
>
> We thank the reviewer for their valuable feedback, we address the questions below. Tables and Figures are provided at ****[Rebuttal Page click here](https://lively-dune-08c51d610.6.azurestaticapps.net/).****
>
> 1. A primary concern is the authors' use of memorization scores precomputed by Feldman & Zhang (2020) while employing different model architectures .... scores.
>
> **A**:  We agree with the reviewer. Feldman & Zhang used Inception for CIFAR100 and ResNet50 for ImageNet. We have added ResNet50 (same as Feldman & Zhang) and Inception (for CIFAR100 same as Feldman & Zhang) in Table 1 (rebuttal page above) along with more baselines (we reran Inception model, due to a minor bug fix, and present the updated scores). The choice of ResNet18 was dictated by its use in prior proxy such as curvature [3].  Additionally, we have included results using multiple architectures in Appendix B.2. These were already part of the appendix but not clearly referenced in the main text; we will fix this in the revised version. Our findings are inline with [3, 4, 5, 6] which have shown that memorization is often a dataset-level property, *given the model has sufficient capacity*. Also, please see our response to fZEA Q3.
>
> 2. Furthermore, ... Without explicitly comparing against such straightforward alternatives, the paper fails to convincingly demonstrate the unique value of CSL over simpler approaches.
>
> **A**:  We have provided the final sample loss, loss sensitivity and forgetting frequency as a baselines (also suggested by reviewer fZEA) in Table 1 (rebuttal page above) . As noted by reviewer woPj, we have also added *Pearson correlation* $\in (-1, 1)$, any magnitude > 0.25 is considered statistically significant correlation. And clearly we see $CSL$ has statistically significant result generally and outperforms other methods. Also, please see our response to reviewer fZEA Q3.
>
> 3. Lemma 5.1, which states .. weight matrix configurations and with no bias.
>
> **A**:  We would like to clarify that Lemma 5.1 indeed holds for a general feed forward neural networks. While the proof in Appendix C.1 begins with the case of a linear layer for simplicity, the result extends to arbitrary feed forward networks (see steps eq. 23 -> 24 this is holds for a general feed forward NN).
>
> Regarding bias, this is a notational simplification. The inclusion of bias can be handled without loss of generality by augmenting the input vector with an additional identity row, a common technique to account for bias. As such, Lemma 5.1 holds for feed forward networks with and without bias. We will revise the appendix to make this generalization more explicit.
>
>  4. The theoretical framework relies on .. assumptions that may not hold in practical deep learning settings .. irregular.
>
> **A**:  First, our theoretical results do not assume convexity. In fact, our analysis applies to widely used unbounded, non-convex, cross-entropy loss, as noted in the paper. Additionally also for any non-convex bounded loss.
>
> Second, the assumptions (e.g., smoothness, stability, bounded gradients) are well-supported in practice. Lipschitz continuity and smoothness of deep networks have been studied and validated in Virmaux & Scaman [1], and uniform stability of SGD has been established by Hardt et al. [2]. Please see our detailed discussion in lines (303-317). Therefore, we believe the theoretical framework remains broadly applicable to standard deep learning practice, especially for networks trained with SGD.
>
> 5. Additionally, the paper is unclear about ... minibatches might affect the derived bounds.
>
> **A**:  Thank you for raising this important point. We understand the potential confusion, and we’d like to clarify. Our proof relies on lemma 5.1, since lemma 5.1 does not depend on the choice of minibatch, our proof can also be extended to SGD with minibatch size larger than 1.
>
> 6. I also don't understand ... data subsets.
>
> **A**:  We would like to clarify that we meant arbitrary **random** subsets. Consider selecting a subset U at random; then T can be set as $\max\{T_{z_i} : z_i \in U\}$. Since Equation (10) holds for such a distribution, we obtain the stated result. We will revise the text to clarify this point and explicitly indicate that the subsets under consideration are random.
>
> 7. Regarding Missing references w.r.t MIA:
>
> **A**: Please see our response to reviewer WoPj Q5.
>
> [1] Virmaux & Scaman Lipschitz regularity of deep neural networks: analysis and efficient estimation. NeurIPS 18\
> [2] Hard  et al. Train faster, generalize better: Stability of stochastic gradient descent. ICML, 16\
> [3] Garg et al. "Memorization Through the Lens of Curvature of Loss Function Around Samples." ICML 24.\
> [4] Garg & Roy. "Samples with low loss curvature improve data efficiency." CVPR 23\
> [5] Lukasik et al. What do larger image classifiers memorise? arXiv preprint arXiv:2310.05337, 2023\
> [6] Ravikumar et al. "Unveiling privacy, memorization, and input curvature links." ICML 24

---

> > ### Comment · Reviewer_ZrxQ · 2025-04-03
> >
> > I thank authors for their thoughtful response and for addressing many of mine and other reviewer's comments, specifically for including suggested baselines and aligning the training procedure with Feldman & Zhang (2020).
> >
> > I'm open to updating my score, but would like to clarify a few points.
> >
> > > (from WoPj rebuttal) This should clarify further how our approach (CSL) integrates with and advances prior research.
> >
> > Can you elaborate on how are you going to update this section?
> >
> > > see steps eq. 23 -> 24 this is holds for a general feed forward NN
> >
> > Can you please elaborate on that? I'm not sure I understand the basis on which you claim eq.24-25 "hold for any deep neural network", when they are derived from eq.22-23, specific to a particular two-layer architecture.

---

> > > ### Author Response · Authors · 2025-04-06
> > >
> > > Thank you for your response, we address your questions below.
> > > 1. Can you elaborate on how are you going to update this section?
> > >
> > > **A:** We were previously limited by the rebuttal characters, below is the updated related work section, we use \<sc\> to denote same citations as in the current version to save space.
> > >
> > > Memorization in deep neural networks has gained attention, with recent works improving our understanding of its mechanisms and implications (\<sc\>). This research is driven by the need to understand generalization (\<sc\>, Kaplun et al., 2022; Bayat et al., 2025), identify mislabeled examples (\<sc\>), and detect out-of-distribution or rare sub-populations (\<sc\> Pezeshki et al., 2023). Additionally, memorization impacts robustness (\<sc\>), unlearning (\<sc\>) and privacy (\<sc\>).
> > >
> > > Privacy is often tested using membership inference attacks (MIA), which tests whether a particular sample was used in training (Shokri et al., 2017; Carlini et al., 2022; Ravikumar et al., 2024b). Learning dynamics has been used in this context to build stronger privacy tests. Liu et al. (2022) leverage loss trajectories and distilled models to improve MIA, while Li et al. (2024) propose SeqMIA, an attention-based recurrent architecture that exploits intermediate model states. Both approaches demonstrate how learning dynamics can reveal training-set membership but at the cost of increased computational overhead. Additionally, Nasr et al. (2021) highlights how mean loss trajectories can reveal privacy leakage under differentially private training establishing a lower bound on leakage identification.
> > >
> > > Learning Dynamics. Beyond privacy, learning dynamics have been studied as proxies for memorization. Mangalam et al. (2019) showed simpler examples are learned first while mislabeled or difficult samples may be repeatedly forgotten or relearned (Toneva et al., 2019; Pleiss et al., 2020; Maini et al., 2022). Jagielski et al. (2023) explored how memorization fades when examples are removed. Carlini et al. (2019a) combine multiple metrics to understand data retention, and Jiang et al. (2021) introduce the C-score to capture hard examples. Other works have proposed loss-sensitivity (Arpit et al., 2017), and sharpness of the loss function (Krueger et al., 2017) as memorization signals. More recently, Garg et al. (2024) used input loss curvature as a proxy for stability-based memorization (Feldman, 2020), supported by theoretical analysis from Ravikumar
> > > et al. (2024a).
> > >
> > > 2. I'm not sure I understand the basis on which you claim eq.24-25 "hold for any deep neural network", when they are derived from eq.22-23, specific to a particular two-layer architecture.
> > >
> > > A:  We provide the proof below. Let's start by considering the a general deep net. We denote:
> > > - The input as $X$,
> > > - The first layer weight matrix as $W_1$, and $\tilde{W}$ is the weights of the entire network (including $W_1$)
> > > - And the intermediate representation (or pre-activation) of the first layer as $IR^{(1)} = W_1 X$.
> > > - Then such a network $f$ can be written as $f(X, \tilde{W}) = g(IR), \quad \text{where} \quad IR = W_1 X \quad \text{Eq 1(R)}$
> > >
> > > > **Condition (1):** The decomposition (Eq 1(R)) holds for any network where the first layer has no skip connection.
> > >
> > > Condition (1) holds for architectures like VGG or ResNet, since they have an initial conv layer without skip a connection. This also holds for ViTs, where input images are projected to patches using an initial linear layer.
> > > For example, in a ResNet, the function $g$ corresponds to all layers after the first convolution.
> > >
> > > Now using the chain rule, the gradient of the loss $\ell$ with respect to the first layer weights $W_1$ can be decomposed into two parts:
> > > 1. The gradient of the loss with respect to the intermediate representation $IR^{(1)}$.
> > > 2. The gradient of the intermediate representation $IR^{(1)}$ with respect to $W_1$.
> > >
> > > Thus, we write:
> > > $\nabla_{W_1} \ell = \nabla_{IR^{(1)}} \ell \cdot \nabla_{W_1} IR^{(1)} = \nabla_{IR^{(1)}} \ell \cdot X^T$ (Eq. 2R)
> > >
> > > Now using the chain rule, we can write the same for input grad:
> > > $\nabla_{X} \ell = \nabla_{IR^{(1)}} \ell \cdot \nabla_{X} IR^{(1)} = W_1^T \cdot \nabla_{IR^{(1)}} \ell$ (Eq. 3R)
> > >
> > > Multiplying by $W_1^T$ on the left for (Eq. 2R) and $X^T$ on the right for (Eq. 3R) gives:
> > > $W_1^T \nabla_{W_1} \ell = \nabla_X \ell \, X^T.$
> > >
> > > Thus we get the lemma 5.1 for any general neural net that satisfies *Condition (1)* above.
> > >
> > > Additionally note: $||W^T_1|| \leq ||\tilde{W}^T||$  and $||\nabla_{W_1} \ell|| \leq ||\nabla_{\tilde{W}} \ell ||$ since $\tilde{W}$ includes $W_1$.
> > > We have
> > > $||\nabla_{X} \ell|| \leq \cfrac{|| W_1^T || \cdot || (X)^+ ||}{s_P} \cdot || \nabla_{W_1} \ell ||$
> > >
> > > Thus:
> > > $||\nabla_{X} \ell|| \leq \cfrac{|| \tilde{W}^T || \cdot || (X)^+ ||}{s_P} \cdot || \nabla_{\tilde{W}} \ell ||$
> > >
> > > We will clarify above condition and proof in the revised version.
> > > We are happy to clarify any further details.

---

### Decision · Program_Chairs · 2025-05-01

**Decision:**

Accept (poster)

**Comment:**

This work proposes a new proxy for measuring memorization, namely using the Cumulative Sample Loss (CSL) that represents the loss of a sample accumulated over the entire training process. The method is less computationally expensive than prior approaches. The important aspect is the theoretical contribution to the analysis of memorization in supervised learning. All the Reviewers agree that this is an important contribution to our community.

The authors put a substantial effort into the rebuttal and addressed most of the Reviewers' comments. However, the empirical analysis is still lacking and more comparisons to baselines should be added. Additionally, the authors should add the missing citations that Reviewers mentioned. Overall, we expect the authors to update the paper according to the feedback from the Reviewers.